# Nuclear genomic signals of the 'microturbellarian' roots of platyhelminth evolutionary innovation

Christopher E Laumer[1]*, Andreas Hejnol[2], Gonzalo Giribet[1]

[1]Museum of Comparative Zoology, Department of Organismic and Evolutionary Biology, Harvard University, Cambridge, United States; [2]Sars International Centre for Marine Molecular Biology, University of Bergen, Bergen, Norway

**Abstract** Flatworms number among the most diverse invertebrate phyla and represent the most biomedically significant branch of the major bilaterian clade Spiralia, but to date, deep evolutionary relationships within this group have been studied using only a single locus (the rRNA operon), leaving the origins of many key clades unclear. In this study, using a survey of genomes and transcriptomes representing all free-living flatworm orders, we provide resolution of platyhelminth interrelationships based on hundreds of nuclear protein-coding genes, exploring phylogenetic signal through concatenation as well as recently developed consensus approaches. These analyses robustly support a modern hypothesis of flatworm phylogeny, one which emphasizes the primacy of the often-overlooked 'microturbellarian' groups in understanding the major evolutionary transitions within Platyhelminthes: perhaps most notably, we propose a novel scenario for the interrelationships between free-living and vertebrate-parasitic flatworms, providing new opportunities to shed light on the origins and biological consequences of parasitism in these iconic invertebrates.

*For correspondence: claumer@oeb.harvard.edu

**Competing interests:** The authors declare that no competing interests exist.

## Introduction

The true flatworms (Platyhelminthes) are one of the major phyla of invertebrate animals, the significance of which may be practically measured in terms of their species diversity and body-plan disparity, as well as from a more theoretical perspective by their role in broader-scale discussions of metazoan phylogeny and as models of fundamental concepts in developmental and stem cell biology, parasitology, and invertebrate zoology. As small acoelomate animals, the free-living members of this phylum ('turbellaria') almost without exception rely on their fully ciliated, non-cuticularized epidermis for all locomotory, respiratory, and circulatory functions, fundamentally constraining them to protected aquatic or humid habitats (*Hyman, 1951*). Despite this restriction, they have successfully radiated in almost all marine and continental aquatic habitats and many humid terrestrial settings, today numbering perhaps tens of thousands of free-living species (*Appeltans et al., 2012*; *Tyler et al., 2012*), of which about 6500 are currently described. The acoelomate condition of Platyhelminthes, among other traits (e.g., their blind gut), has also historically positioned them prominently as figures of supposedly 'primitive' Bilateria. While molecular phylogenetics has for over a decade nested this taxon well within the protostome clade Spiralia (*Carranza et al., 1997*; *Baguñà and Riutort, 2004*), displacing them from their classical position as early-branching bilaterians, modern manifestations of the debate over the relevance of such characters continue, with the role of acoelomate early-branching bilaterians (but see *Philippe et al., 2011*) being taken over by Xenacoelomorpha (*Hejnol et al., 2009*; *Srivastava et al., 2014*), themselves formerly Platyhelminthes. This fragmentation of the phylum is not, however, fully incompatible with the classical interpretation of the 'primitive' nature of some aspects of platyhelminth organization, and indeed interest in this debate is resurging with, for example, recent molecular

**eLife digest** Flatworms are relatively simple invertebrates with soft bodies. They can be found living in nearly every aquatic environment on the planet, are well-known for their ability to regenerate, and some species live as parasites in humans and other animals. Studies of the physical characteristics of flatworms have provided us with clues about how some groups, for example, the parasitic flatworms, have evolved, but the evolutionary origins of other groups of flatworms are less clear.

The genetic studies of flatworm evolution have focused on a single gene that makes a molecule called ribosomal ribonucleic acid, which is required to make all the proteins in flatworms and other animals. By comparing the sequences of this gene in different species of flatworm, it is possible to infer how they are related in evolutionary terms—that is, species with shared gene sequence features are likely to be more closely related than two species with less similar gene sequences. Although this approach has proved to be useful, it has also produced some results that conflict with the conclusions of previous studies.

Here, Laumer et al. studied the evolution of flatworms using an approach called RNA sequencing. This approach made it possible to sequence many hundreds of genes in all major groups of flatworms, and compare these genes in different species. Laumer et al. used the data to build a 'phylogenetic tree' that infers the evolutionary relationships between the different groups of flatworms.

This tree provides evidence that supports some of the ideas about flatworm evolution produced by the previous studies based on both physical features and ribosomal ribonucleic acid. It also presents several unexpected evolutionary relationships; for example, it suggests that the parasitic flatworms are most closely related to a group of small flatworms called Bothrioplanida, which are predators of other invertebrates. Bothrioplanida can live in many freshwater environments, and the physical characteristics that allow them to survive might resemble those found in the earliest parasitic flatworms.

The phylogenetic tree made by Laumer et al. represents a guide for researchers seeking clues to the origins of the genetic and developmental innovations that underlie the various physical features found in different flatworms.

phylogenetic evidence for the paraphyly of 'Platyzoa' (an assemblage of small acoelomate and pseudocoelomate spiralians including Platyhelminthes, Gastrotricha, and Gnathifera [*Struck et al., 2014*; *Laumer et al., 2015*]).

Irrespective of the broader evolutionary implications of pan-platyhelminth characteristics, the clade is also widely known for those of its members which have been adopted as models of fundamental zoological concepts. Freshwater planarians such as *Schmidtea mediterranea* (Tricladida) have a long history of utility in classical zoology, and modern molecular genetic appropriations of this system, as well as the more recently developed model *Macrostomum lignano* (Macrostomorpha) (*Ladurner et al., 2005*), have provided insights into especially non-embryonic developmental processes inaccessible in other familiar invertebrate models, such as whole body regeneration (*Sánchez Alvarado, 2012*), stem-cell maintenance (*Sánchez Alvarado and Kang, 2005*), tissue homeostasis (*Pellettieri and Alvarado, 2007*; *Reddien, 2011*), and aging (*Mouton et al., 2011*). The marine polyclad flatworms (Polycladida) have also been a subject of perennial study, not least due to their compelling reproductive biology: although they engage in (an often elaborately achieved [*Michiels and Newman, 1998*]) internal fertilization unlike most other marine macro-invertebrates, their embryos show a clear quartet spiral cleavage and cell fate (*Boyer et al., 1998*), and many species present a long-lived planktotrophic larva (*Rawlinson, 2014*) with well-developed ciliary bands and cerebral ganglia, which have been homologized to the trochophora larvae of other Spiralia (*Nielsen, 2005*). Furthermore, polyclads, due to their large clutch sizes, endolecithal yolk (*Laumer and Giribet, 2014*), and thin eggshells, represent the only platyhelminth lineage in which experimental manipulation of embryonic development is possible. Lastly, but far from least, platyhelminths have been long considered masters of parasitism (*Kearn, 1997*). Although nearly all 'turbellarian' lineages evince some symbiotic representatives (*Jennings, 2013*), the flatworm knack

for parasitism reaches is zenith in a single clade, Neodermata (*Ehlers, 1985*). Indeed, the obligate vertebrate parasitism manifested by this group of ecto- and endoparasitic flukes (Polyopisthocotylea, Monopisthocotylea, Digenea, and Aspidogastrea) and tapeworms (Cestoda) is perhaps the single most evolutionarily successful adoption of a parasitic habit in the animal kingdom (in contrast to the case of the nematodes, in which vertebrate parasitism has multiple evolutionary origins [*Dieterich and Sommer, 2009*]). Central among the adaptations responsible for the success of Neodermata—reflected in its some 40,000–100,000 estimated species (*Rohde, 1996*; *Littlewood, 2006*)—was the invention (among other synapomorphies [*Littlewood, 2006*; *Jennings, 2013*]) of the eponymous 'neodermis', a syncytial tegument which plays specialized roles in host attachment, nutrient appropriation, and immune system evasion (*Tyler and Tyler, 1997*; *Mulvenna et al., 2010*). The neodermis has intimately (and ostensibly, irreversibly [*Littlewood, 2006*]) tied the evolutionary success of this lineage to that of its hosts, and as a result, neodermatans appear to have outstripped the diversification of their free-living ancestors by nearly an order of magnitude, with evidence that most vertebrate species (not to mention many species of intermediate hosts from diverse animal phyla) are infected by at least one neodermatan flatworm (*Poulin and Morand, 2000*; *Littlewood, 2006*), sometimes with startling host specificity (particularly in monogenean trematodes). Human beings and their domesticated animals have also not escaped the depredations of neodermatans, which include the etiological agents of several diseases of profound incidence, morbidity, and socioeconomic impact (*Berriman et al., 2009*; *Torgerson and Macpherson, 2011*; *Tsai et al., 2013*), such as schistosomiasis (*Gryseels et al., 2006*), the second-most globally important neglected tropical disease (after malaria), affecting almost 240 million people worldwide.

Despite their scientific preeminence, however, planarians, polyclads, and neodermatans remain merely the best-known branches of a much larger and deeper phylogenetic diversity of platyhelminths (*Hyman, 1951*; *Karling, 1974*; *Rieger et al., 1991*). Indeed, these three lineages are among the only flatworms to exhibit large (>1–2 mm) body size; accordingly, the 9–10 other flatworm orders are usually collectively referred to as 'microturbellarians', a practical term acknowledging their shared, albeit plesiomorphic, adaptations to interstitial habitats (*Giere, 2015*). No one microturbellarian taxon shows the remarkable regenerative capacity of some triclad species (*Egger et al., 2007*), nor the clear, experimentally accessible spiral cleavage of polyclads (*Martín-Durán and Egger, 2012*), nor the profound commitment of neodermatans to parasitic habits (*Jennings, 2013*), but several taxa do exhibit lessened or modified versions of some or all of these traits. Understanding the broader evolutionary significance and initial emergence of these emblematic flatworm traits, therefore, requires phylogenetically constrained comparisons between these familiar taxa and their relatively obscure 'microturbellarian' relatives.

To this end, the internal phylogeny of Platyhelminthes has gained much clarity in recent years through the analysis of rRNA sequence data (*Littlewood et al., 1999*; *Lockyer et al., 2003*; *Baguñà and Riutort, 2004*; *Littlewood, 2006*; *Laumer and Giribet, 2014*), for instance via the demonstration of the polyphyly of taxa such as Seriata (Tricladida, Proseriata, and Bothrioplanida; [*Sopott-Ehlers, 1985*]) and Revertospermata (Fecampiida and Neodermata; [*Kornakova and Joffe, 1999*]), as well as through support for some classically defined scenarios such as the sister-group relationship between Catenulida and Rhabditophora (*Larsson and Jondelius, 2008*), and more recently, evidence for Macrostomorpha (Haplopharyngida+Macrostomida) and the early-branching position of 'lecithoepitheliates' within a clade of ectolecithal flatworms (Euneoophora; [*Laumer and Giribet, 2014*]). Nonetheless, even with these taxon-rich data sets, several key deep phylogenetic splits remain lacking in resolution. While it is clear that polyclads occupy a relatively basally branching position within Rhabditophora, their relationship to Macrostomorpha and possibly Prorhynchida remains unclear. Similarly, though rRNA support for a clade called Adiaphanida (comprising Tricladida, Fecampiida, and Prolecithophora; [*Norén and Jondelius, 2002*]) is nearly unequivocal, the internal relationships within this clade remain poorly supported. Finally, and most importantly, while rRNA-based phylogenies have proven effective in falsifying previous hypotheses on the origins of Neodermata—perhaps the most intensely researched goal of platyhelminth systematics (*Baguñà and Riutort, 2004*; *Littlewood, 2006*)—to date they have not been successful in producing practically useful, well-supported alternative hypotheses. Instead, available phylogenies have largely indicated that Neodermata has no close relationship with any 'turbellarian' order: its sister group is currently understood to be a diversified clade consisting of Tricladida, Prolecithophora, Fecampiida, Rhabdocoela, and perhaps Proseriata (*Littlewood et al., 1999*; *Lockyer et al., 2003*; *Baguñà and Riutort, 2004*; *Littlewood, 2006*;

*Laumer and Giribet, 2014*), thereby complicating comparisons between free-living flatworms and Neodermata, and implying an ancient origin of obligate vertebrate parasitism in the phylum.

To test the so far almost entirely rRNA-based molecular phylogeny of Platyhelminthes, we conducted an RNA-seq survey encompassing representatives of all orders of 'turbellarian' flatworms and related spiralian outgroups (*Supplementary file 1*). Such data represent a potent source of protein-coding sequence data suitable for phylogenetic analysis (*Dunn et al., 2013*; *Lemmon and Lemmon, 2013*; *Yang and Smith, 2014*), and given the much smaller target size and the comparative dearth of complex repeat landscapes in transcriptome data, may be assembled de novo with much less required sequencing depth and computational difficulty than genome data (*Hittinger et al., 2010*; *Haas et al., 2013*), despite the complicating phenomena of alternative splicing and wide variance in gene expression levels. From such de novo assemblies, we have selected a single representative open reading frame per putative unigene per species, and employed a sensitive, graph-based orthology assignment algorithm (*Roth et al., 2008*) to organize these peptides, together with gene models derived from annotated genomes, into putatively orthologous sets. These ortholog groups were then subjected to multiple sequence alignment, with masking of poorly aligned regions, and a subset of 516 such groups were selected for phylogenetic analysis using an automatic matrix reduction procedure (*Misof et al., 2013*). We constructed a supermatrix representing a concatenation of this subset, composed of 132,299 amino acids (with 48.58% matrix completeness) which was analyzed under site-heterogeneous models (*Lartillot and Philippe, 2004*; *Le et al., 2012*) using both maximum likelihood (ML) (*Stamatakis and Aberer, 2013*) and Bayesian inference (BI) approaches (*Lartillot et al., 2013*) (*Figure 1*). For these concatenated analyses, we also employed several approaches to control for systematic errors, for example, by trimming sites that fail tests of compositional heterogeneity (*Foster, 2004*; *Criscuolo and Gribaldo, 2010*) or by leveraging models built to control the effects of heterotachous substitution (*Philippe et al., 2005*; *Pagel and Meade, 2008*). We also considered phylogenetic signal from a gene-tree centric perspective, inferring individual ML trees for each gene, and summarizing the predominant (and sometimes, conflicting; [*Fernández et al., 2014*]) splits in this set of unrooted, incomplete gene trees using both quartet supernetworks (*Grunewald et al., 2013*) (*Figure 2*) and an efficient species-tree algorithm (*Mirarab et al., 2014*) (*Figure 3*). Such approaches may mitigate the inter-gene heterogeneity in branch length and amino acid frequency introduced by concatenation (*Liu et al., 2015*), albeit at the cost of introducing a greater sampling error into gene-tree estimation (a cause of apparent gene-tree incongruence perhaps more prevalent at this scale of divergence than the genuine incongruence modeled by most species-tree approaches, namely incomplete lineage sorting). We also performed taxon deletion experiments to test for the effects of long-branch attraction in influencing the placement of the fast-evolving Neodermata within the phylogeny (*Figures 4, 5*). Considered together, our analyses provide a consistent signal of deep platyhelminth interrelationships, demonstrating a combination of groupings familiar from the eras of classical morphological systematics and rRNA phylogenetics, as well as several novel but nonetheless well-supported clades, whose provenance and broader evolutionary significance we now consider (*Figure 6*).

## Results and discussion

### Monophyly and outgroup relationships of Platyhelminthes

Platyhelminthes, in its modern conception, is comprised of two major clades, Catenulida and Rhabditophora, each themselves morphologically well-defined, which however do not share any known morphological apomorphies (*Ehlers, 1985*; *Smith et al., 1986*). Nonetheless, in rRNA phylogenies to date (*Larsson and Jondelius, 2008*), as well as in the present analyses (*Figures 1*), the monophyly of Platyhelminthes finds nearly unequivocal support. The precise position of the phylum within Spiralia remains controversial, though recent studies have argued for a sister-group relationship with Gastrotricha within a paraphyletic 'Platyzoa' (*Struck et al., 2014*; *Laumer et al., 2015*). As we intended only to resolve relationships within Platyhelminthes, our outgroup sampling is insufficient to test the status of Platyzoa, as we lack more distant outgroups to Spiralia (members of Ecdysozoa). Nonetheless, in all our analyses, our sampled platyzoan taxa fall between Platyhelminthes and our representatives of Trochozoa (Annelida and Mollusca), indicating either mono- or paraphyly of this taxon (*Struck et al., 2014*; *Laumer et al., 2015*). It is, however, interesting to note the comparatively long branch distance separating Catenulida and Rhabditophora, which may imply that future efforts to test the placement of

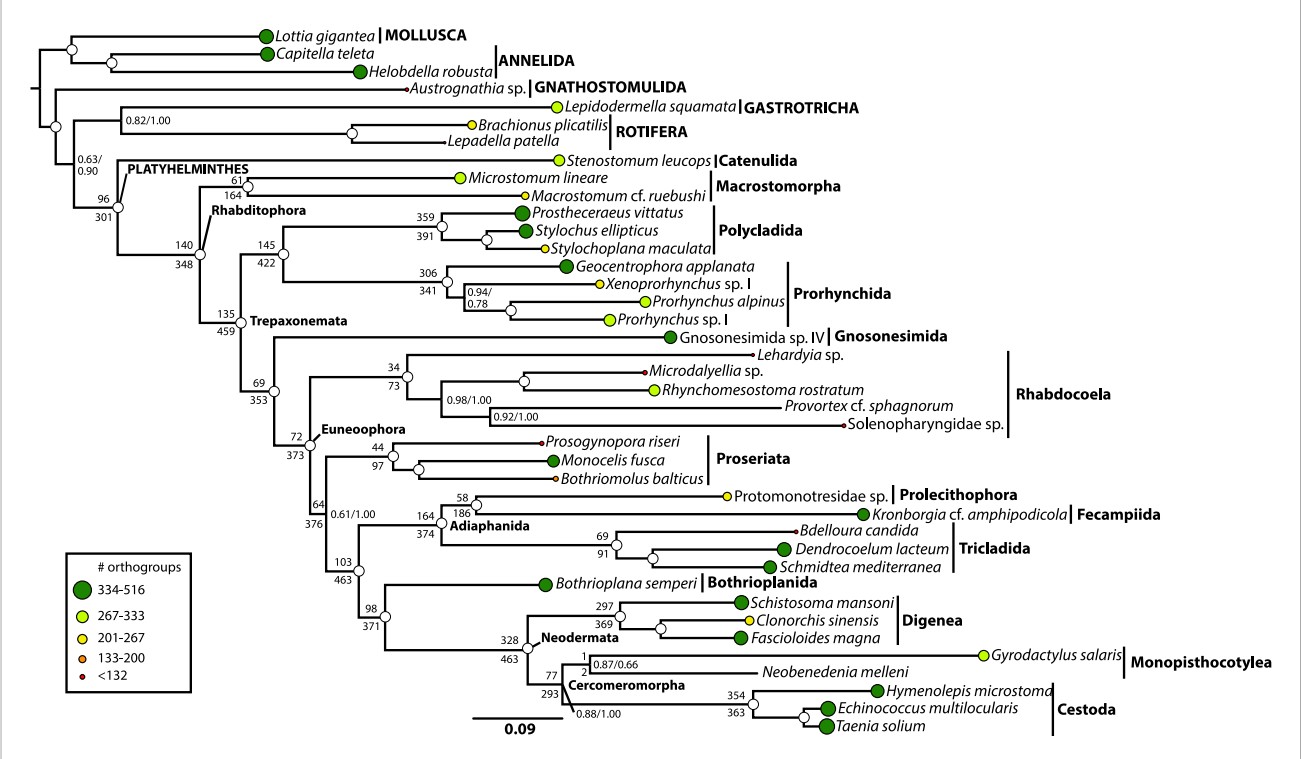

**Figure 1**. Phylogenetic relationships of Platyhelminthes, encompassing 25 'turbellarian' species, 8 representatives of Neodermata, and 7 spiralian outgroups. Phylogram represents results from a maximum likelihood (ML) analysis of 516 predicted orthogroups (120,527 aligned amino acid sites trimmed of nonstationary and poorly aligned residues), analysed in ExaML v1.0.0 under LG4M+F (from which the phylogram shown was taken), and also in phyML (v20130927) under LG+FR4+F+IL, with support values to the right of each node representing clade frequency in 100 bootstraps or aBayes supports, respectively (complete support in both measures indicated by a white dot). Values below internodal edges represent the number of genes available to test each clade (decisive genes); values above such edges represent the percentage of these genes whose ML trees are consistent with this clade (congruence frequency). Colored circles at tips of tree are given with diameter in proportion to the number of orthogroups represented for that taxon in the supermatrix, and colored as per the legend in the lower left.

The following figure supplements are available for figure 1:

**Figure supplement 1**. Maximum likelihood phylogram resulting from analysis of untrimmed 516-gene matrix.

**Figure supplement 2**. Majority rule consensus (MRC) phylogram of a Bayesian Markov Chain Monte Carlo sampling with the CAT+GTR+Γ4 model in PhyloBayes-MPI v1.4e.

Platyhelminthes within Spiralia would do well to sample Catenulida, if long-branch attraction artifacts are to be avoided.

## Interrelationships among free-living taxa of Rhabditophora
### Macrostomorpha is the sister group of the remaining Rhabditophora
Recent molecular phylogenies (*Lockyer et al., 2003*; *Baguñà and Riutort, 2004*; *Laumer and Giribet, 2014*) agree with classical hypotheses (*Ehlers, 1985*) in placing the orders Macrostomorpha and Polycladida—which together with Catenulida comprise the endolecithal flatworms, viz. those which do not produce distinct yolk-bearing cells—as early-diverging lineages within Rhabditophora. However, support for the precise branching order of these early rhabditophoran divergences has remained elusive, with some analyses placing polyclads as the earliest-diverging rhabditophorans, others with macrostomorphs in this position (*Laumer and Giribet, 2014*), and still others suggesting a clade of both taxa (sometimes also including the lecithoepitheliate order Prorhynchida; [*Lockyer et al., 2003*]). All our analyses position our sampled Macrostomorpha as the earliest-branching rhabditophoran lineage (*Figures 1–5*), to our knowledge rendering this the first time this topology

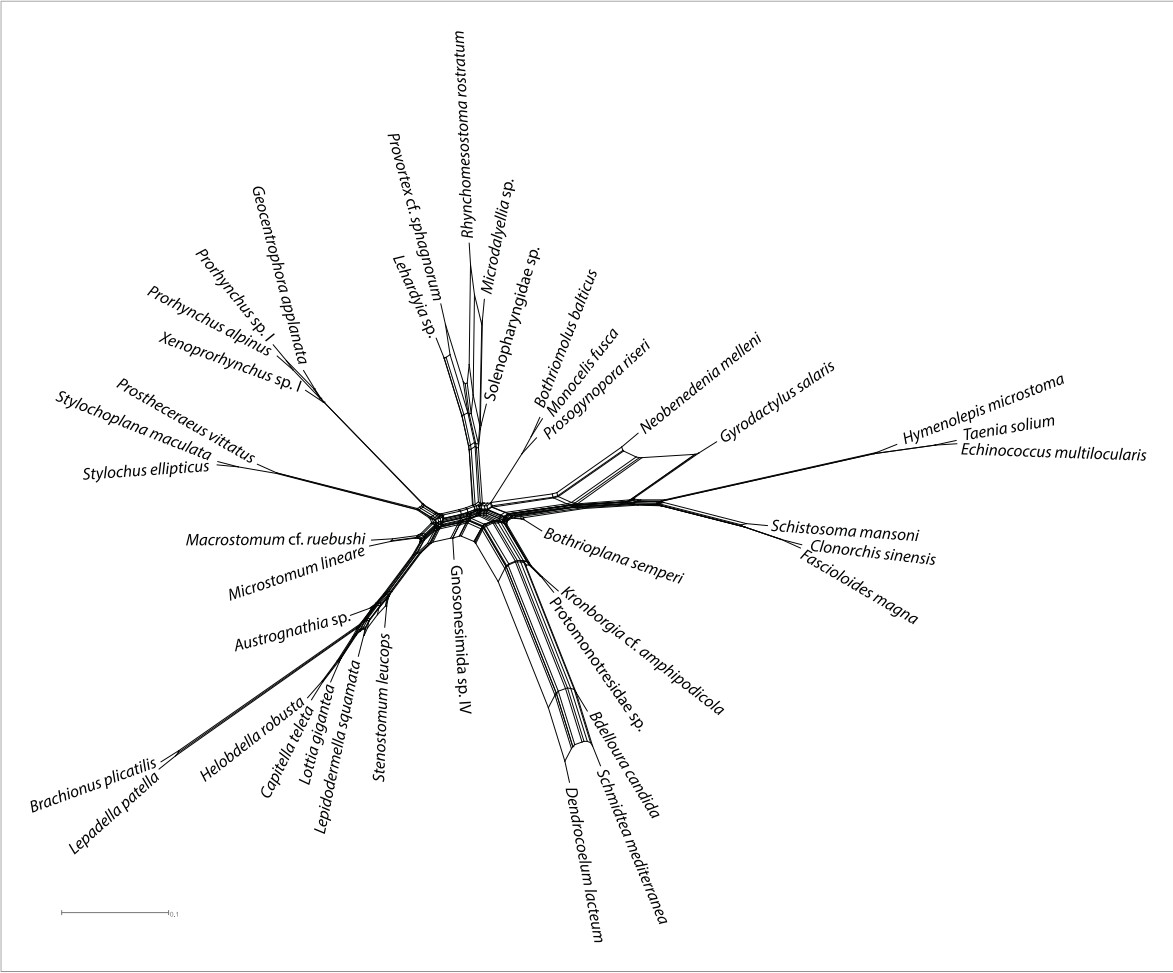

**Figure 2**. Quartet supernetworks built from 516 individual ML gene trees, showing predominant inter-gene conflicts in genes selected for concatenation. A qualitatively nearly identical topology (not shown) was recovered using bootstrap majority rule consensus trees as input. Edge weights were calculated in SuperQ v1.1, with the 'balanced' linear objective function, and no filter applied. Supernetwork was visualized in SplitsTree v4 using default settings. DOI: 10.7554/eLife.05503.006

has been recovered by a molecular data set with strong branch support. Interestingly, this topology is also consilient with a classical hypothesis proposed on the basis of a convincing ultrastructural synapomorphy: the clade comprising the non-macrostomorph Rhabditophora has been named Trepaxonemata, in recognition of the fact that all representatives of this clade (occasional reversals notwithstanding; [*Jondelius et al., 2001*; *Justine, 2001*]) appear to have a '9 + 1' arrangement of microtubule fibers in the core axoneme in mature spermatozoa, with the central microtubule element formed into a spiral (*Ehlers, 1985*; *Justine, 2001*).

## Polycladida is closely related to Prorhynchida, rendering Lecithoepitheliata non-monophyletic

Polycladida and Prorhynchida represent among the best-represented groups in our data set both in terms of taxon sampling and sequencing depth (*Supplementary file 1*), and the high support we observe for the placement of both taxa is therefore unsurprising. Our recovered sister-group relationship of these taxa is (*Figures 1–6*), however, unexpected, at least at first glance. Traditionally, Prorhynchida has been grouped with the order Gnosonesimida in a clade called Lecithoepitheliata, reflecting the hypothesis that these taxa both present a primitive form of ectolecithality, an appropriation of oocyte functions such as yolk storage and cortical granule synthesis into a novel cell type called the vitellocyte (reviewed thoroughly by *Laumer and Giribet, 2014*). However, recognizing that, aside from gross anatomical similarity in the structure of female gonads, prorhynchids and gnosonesimids share essentially no derived morphological traits, many authors have expressed

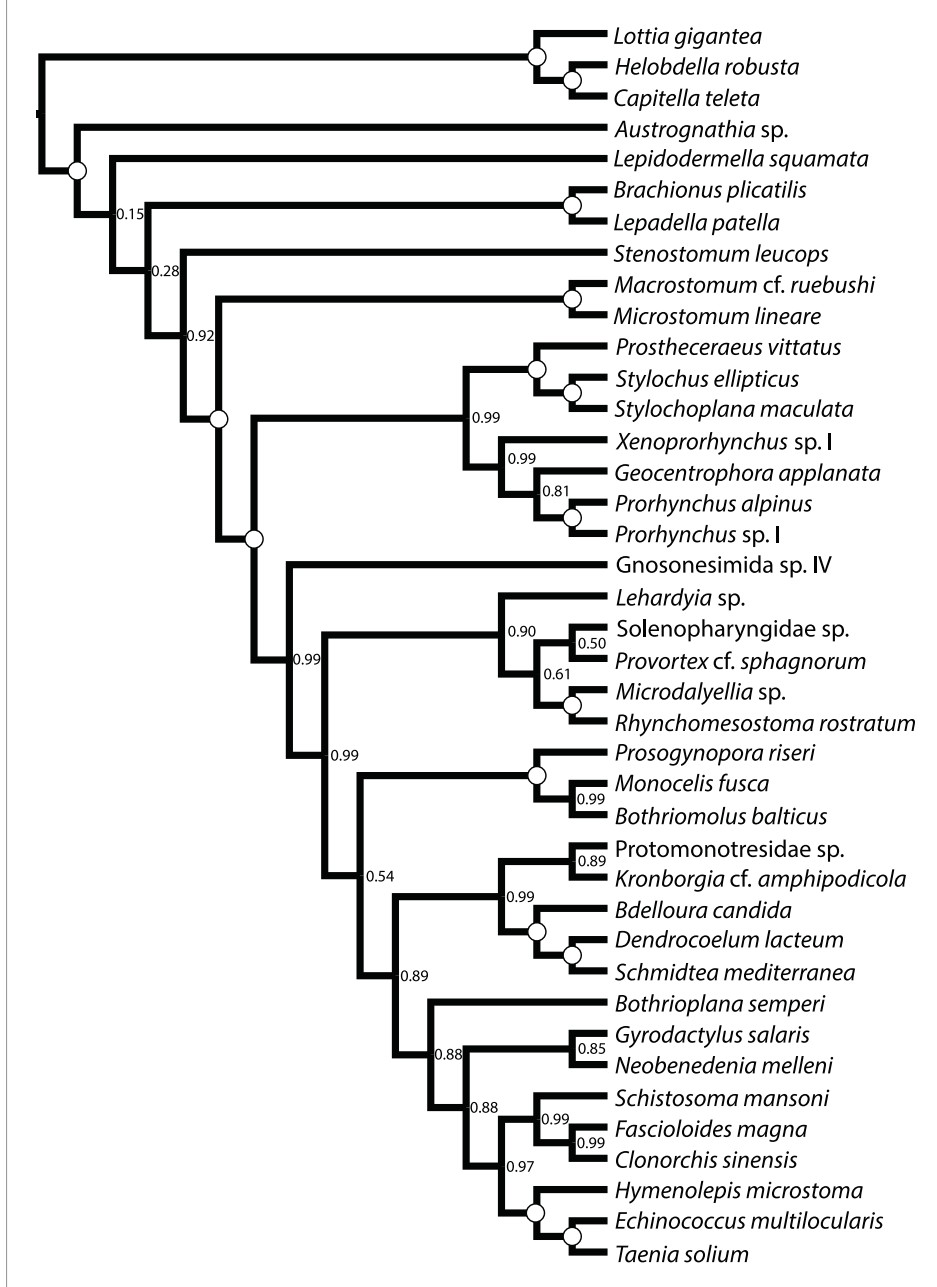

**Figure 3**. ASTRAL species tree. Constructed under default settings from 516 input unrooted partial gene trees inferred in RAxML v8.0.20. Nodal support values reflect the frequency of splits in trees constructed by ASTRAL from 100 bootstrap replicate gene trees using the -b flag; gene- and site-level bootstrapping (-g) was not performed.

skepticism of the Lecithoepitheliata hypothesis (**Karling, 1968**; **Martens and Schockaert, 1985**; **Timoshkin, 1991**). In the first study to thoroughly sample molecular data from representatives of Gnosonesimida and Prorhynchida, **Laumer and Giribet (2014)** found support for lecithoepitheliates as a clade or a paraphyletic grade (depending on mode of analysis) closely related to a clade comprising all other ectolecithal flatworms (Euneoophora). The present RNA-seq-based phylogeny, however, implies non-monophyly of Lecithoepitheliata, with Gnosonesimida more closely related to Euneoophora than to Prorhynchida (**Figure 1**). Although the present study contradicts Laumer and Giribet's (**Laumer and Giribet, 2014**) rRNA-based placement of Prorhynchida, we note that the position of at least Gnosonesimida as sister to Euneoophora is in fact in accordance with

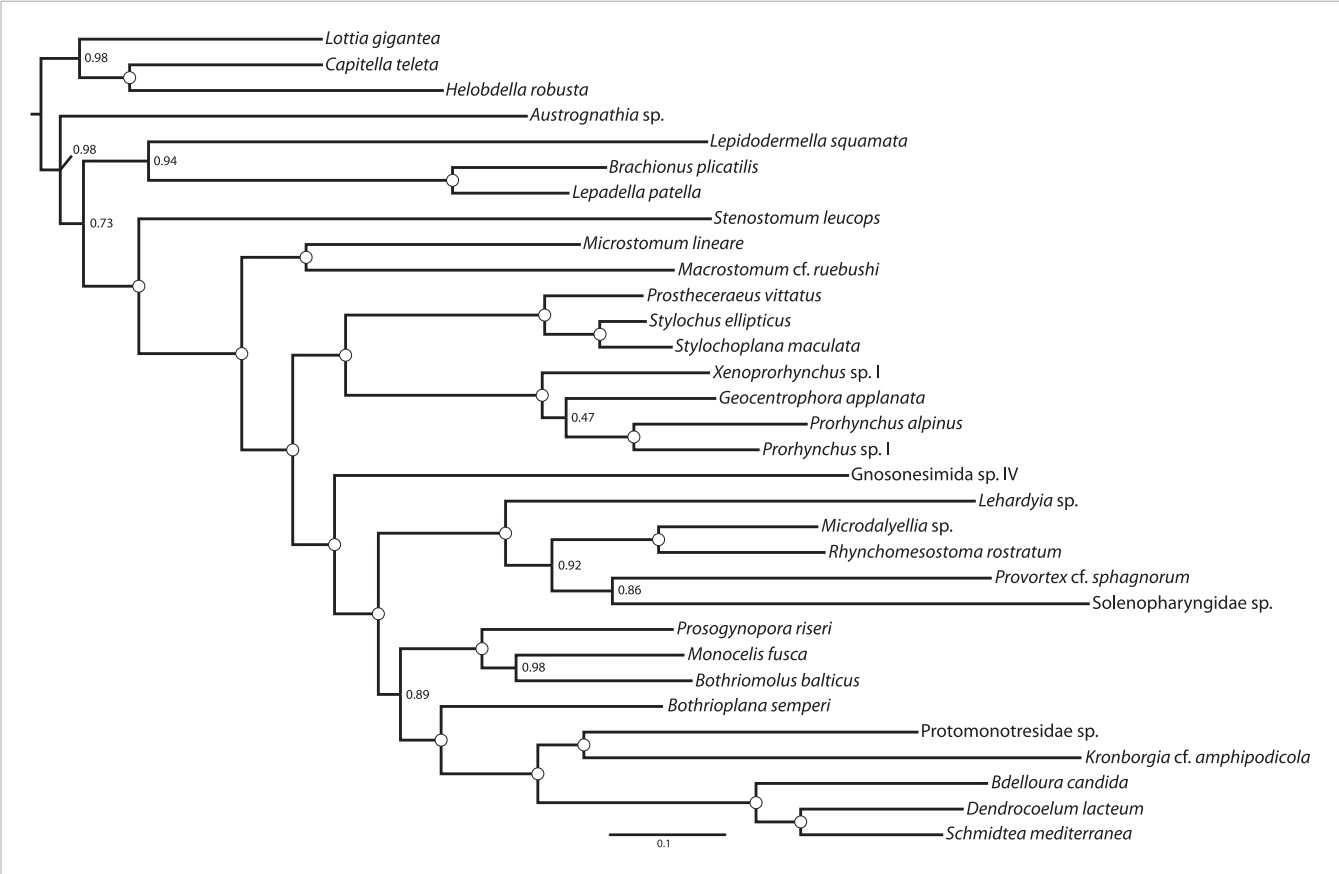

**Figure 4**. ML phylogram inferred from a version of the BMGE-trimmed matrix in which all taxa of Neodermata have been deleted. Tree inferred in ExaML v1.0.0 under the LG4M+F model; nodal support values represent the frequency of splits in 100 bootstrap replicates.

evidence from rRNA, supporting the proposition of a stepwise, if not necessarily single, origin of ectolecithality.

In contrast, the significance of a Polycladida+Prorhynchida clade for the evolution of flatworm ectolecithality is less clear. One possible interpretation posits an independent origin (and therefore, non-homology) of the vitellocytes produced by members of Prorhynchida compared to those in Gnosonesimida+Euneoophora, and indeed, it has been pointed out that on the ultrastructural level gnosonesimid and prorhynchid female gonads ('germovitellaria') show little similarity (**Bogolyubov and Timoshkin, 1993**). Another reading of this topology, however, would retain the homology of vitellocytes and the hypothesis of a single, stepwise origin of ectolecithality via a 'lecithoepitheliate' intermediate, further necessitating only its subsequent secondary loss in Polycladida—a scenario that, interestingly, has been argued before on morphological grounds (**Karling, 1967**). Under this topology, phylogenetics alone cannot meaningfully contribute further to discussion on the origins of ectolecithality: if losses are treated equally parsimonious as gains, these two scenarios are indistinguishable. Further insight must therefore be sought from comparative ultrastructural and especially developmental genetic inquiries on oogenesis and vitellocyte specification in representatives of Gnosonesimida, Prorhynchida, Euneoophora and now, Polycladida. The significance of this question is furthermore not limited to specialists on ectolecithality: particularly if polyclads are secondarily endolecithal, this has profound consequences for the interpretation of studies of polyclad development, since ectolecithality is apparently associated with more or less dramatic developmental modifications (**Thomas, 1986**; **Martín-Durán and Egger, 2012**). Indeed, while both taxa retain a recognizable quartet spiral early cleavage and cell lineage, perhaps to a greater extent than any other rhabditophoran flatworms, the early development of Prorhynchida seems in many ways less modified

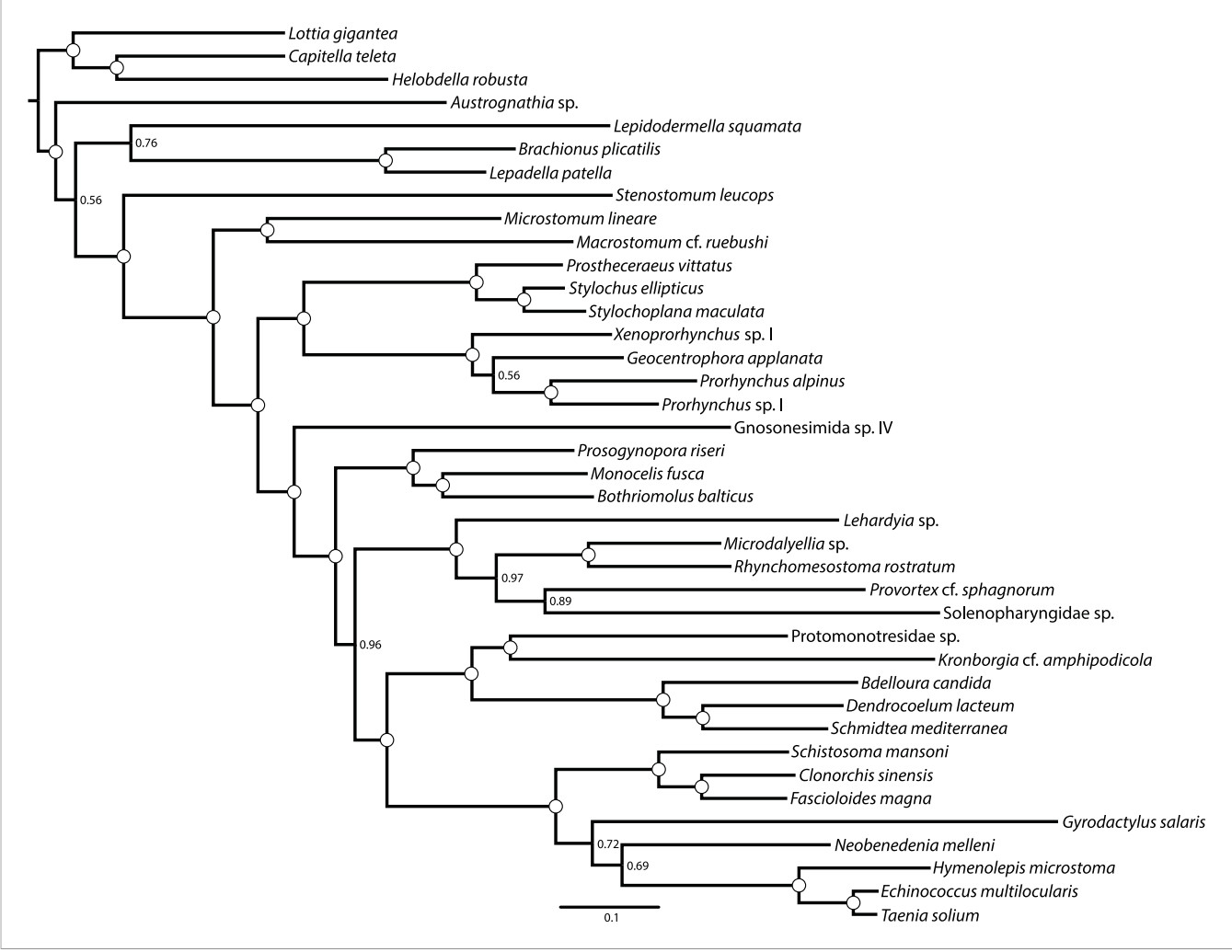

**Figure 5**. ML phylogram inferred from a version of the BMGE-trimmed matrix from which *Bothrioplana semperi* has been deleted. Tree inferred in ExaML v1.0.0 under the LG4M+F model; nodal support values represent the frequency of splits in 100 bootstrap replicates.

from a canonical spiralian cleavage program than that of Polycladida, in which the mesentoblast appears to have shifted assignment from 4*d* to 4*d²*, and in which there is a full degeneration of the fourth quartet macromeres and micromeres 4*a-c* (*Reisinger et al., 1974*; *Boyer et al., 1998*; *Martín-Durán and Egger, 2012*).

In contrast to the question of the homology of ectolecithal oogenesis, however, our analyses can inform on another proposed evolutionary developmental scenario: the interpretation of polyclad larvae (e.g., Götte's and Müller's larvae) as modified trochophores (*Nielsen, 2005*; *Lapraz et al., 2013*). Under our topology (*Figure 6*), for this homology proposal to be true would require at least four independent losses of planktonic larvae within Platyhelminthes alone (in Catenulida, Macrostomorpha, Prorhynchida, and Euneoophora) to say nothing of further additional required losses within 'Platyzoa'. It is thus far more parsimonious to view polyclad larvae as one or more independent acquisitions private to this group (*Rawlinson, 2014*). Altogether, the position of Polycladida recovered in our analyses suggests that the order may be more derived within Platyhelminthes than has been widely appreciated, warranting particular caution in the interpretation of developmental data from the only platyhelminth taxon amenable to experimental embryological research (*Boyer et al., 1998*; *Rawlinson, 2010*; *Lapraz et al., 2013*; *Rawlinson, 2014*).

To our knowledge, a clade of Prorhynchida and Polycladida has not been seriously considered before, raising the question of what apomorphies may unite them. There are, however, numerous

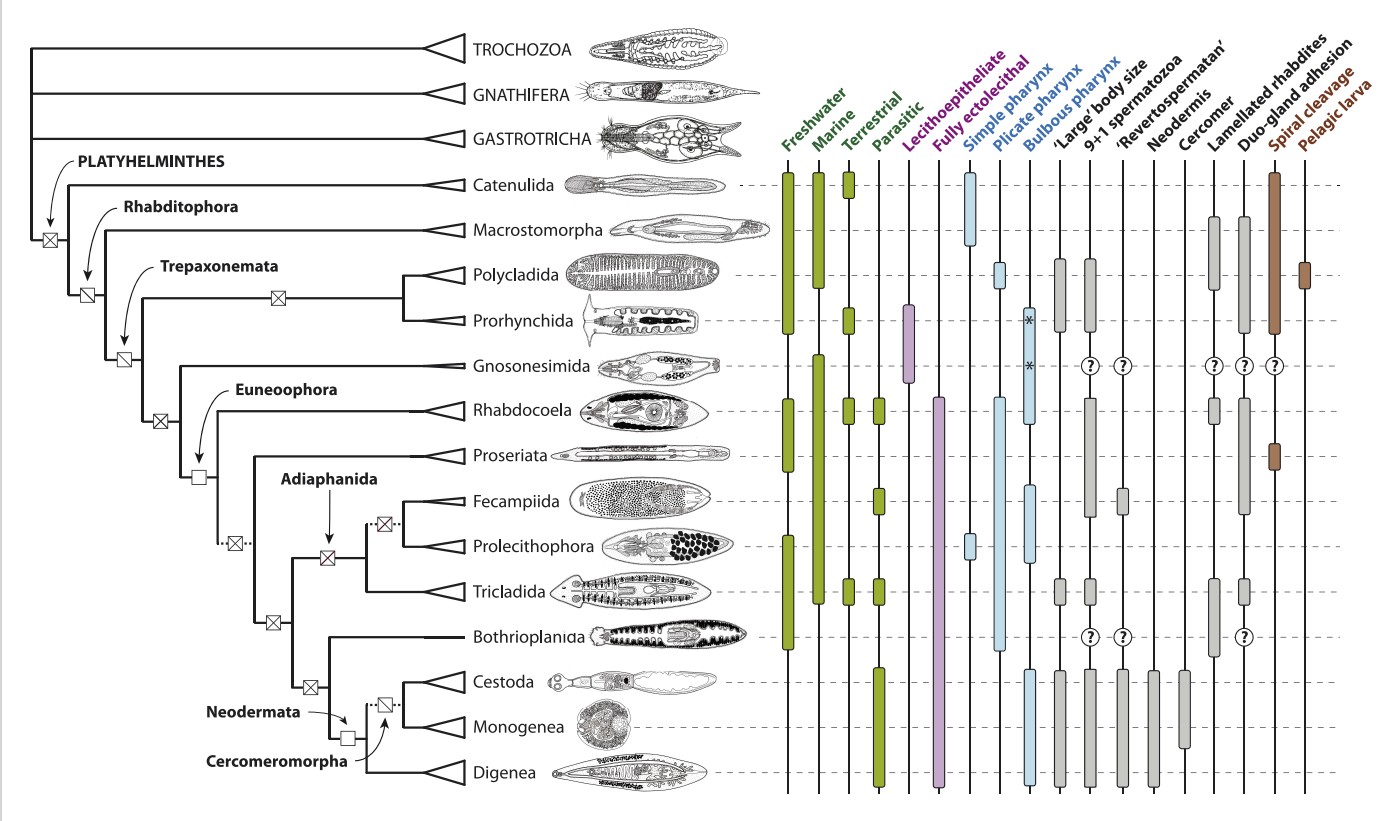

**Figure 6**. Summary depiction of the phylogenetic results presented in the present study. All terminal taxa (turbellarian orders, neodermatan classes, and outgroup clades) are shown with equilateral triangles log-scaled in proportion to described species number (largely following [*Martín-Durán and Egger, 2012*]). Clades deemed to have equivocal or only preliminary support in these analyses are shown with a dashed subtending branch. Named clades discussed in the text are labeled. Clades with known homoplasy-free synapomorphies are labeled with an open square, whereas those with proposed synapomorphies showing inferred losses are labeled with a single diagonal line, and clades without any known synapomorphies are shown with a cross. Traits commonly referred to in discussions of platyhelminth phylogenetics are given to the right of the phylogeny; terminal taxa are shown filled in if any representatives have been reported in published studies to manifest the trait in question (although numerous reversals apparently occurring subsequent to the origin of a clade are not depicted), and clades for which no published observations are available are labeled with a question mark. The orders Prorhynchida and Gnosonesimida are given with a '*' in the 'Bulbous pharynx' column to indicate that although the pharyngeal types manifested in these orders have been described under separate names (variable and coniform pharynx, respectively), both represent, by definition, a bulbous pharynx.

Line drawings in Figure 6 modified with permission from the following sources:

Gastrotricha (unspecified chatonotoid) – BIODIDAC database (http://biodidac.bio.uottawa.ca/)

Gnathostomulida (Gnathostomula peregrina) – *Kirsteuer, 1969*

Annelida (Hirudinea sp.) – BIODIDAC database (http://biodidac.bio.uottawa.ca/)

Prorhynchida (after Geocentrophora marcusi) – original

All others – Images modified from original illustrations provided courtesy of Janine Caira (*Caira and Littlewood, 2013*).

correspondences between at least individual exemplars of these taxa. Several prorhynchid species (e.g., *Geocentrophora applanata*) evince guts with numerous lateral, though non-bifurcating, branches reminiscent of at least smaller polyclads. Moreover, the anatomical construction of the male copulatory system of some polyclads corresponds strongly to the prorhynchid genera *Prorhynchus* and *Xenoprorhynchus*—particularly, many species of Leptoplanoidea present a single sac-like seminal vesicle connected via an intravesicular duct to a strongly muscular vesicula granulorum multiply pierced by many extravesicular glands (*Bulnes, 2010*) and capped with a terminal hypodermic needle-like penis stylet (*Prudhoe, 1985*). Furthermore the relative positioning of mouth, male, and female gonopores in several polyclad taxa resembles the basic reproductive organization of Prorhynchida: particularly several Euryleptoidea have an unruffled, forward-oriented plicate pharynx (though somewhat different in fine structure from the prorhynchid variable pharynx; [*Watson and Rohde, 1992*]) with an anterior mouth and a male pore very closely behind or even fused with the mouth (as in the genus *Chromyella*). Within

Prorhynchida, several little-known species (particularly *Prorhynchus putealis*, *Prorhynchus tasmaniensis*, *Prorhynchus haswelli*, and *Prorhynchus insularis*) present a very large (7+ cm) body size dwarfing even many 'macroturbellarian' polyclads, and most members of Prorhynchida bear prominent anterior auricles similar to those of many cotylean polyclads (e.g., the genus *Boninia*), giving the two groups occasionally very similar habitus. Before these or other characters can be raised as legitimate synapomorphies, however, the internal phylogenies of both orders, but particularly the morphologically highly plastic Polycladida, must be resolved, so as to distinguish plausibly plesiomorphic traits from those derived well within this diverse clade. For this reason, despite the unequivocal signal for this sister-group relationship, we refrain for the moment from recognizing this clade with a higher taxonomic name.

## Rhabdocoela and Proseriata are newly positioned near the base of a clade of fully ectolecithal flatworms, Euneoophora

The two most diverse microturbellarian orders are Proseriata, with about 400 described species (*Curini-Galletti, 2001*), and Rhabdocoela, with about 1530 described species (*Van Steenkiste et al., 2013*), although the diversity of both of these principally marine interstitial taxa remains poorly known. In recent rRNA-based molecular phylogenies, both orders have been recovered as deep branches within a monophyletic Euneoophora, with strong resampling support for a scenario in which Proseriata is the earliest splitting branch, and more modest support for Rhabdocoela as the sister group to Adiaphanida (see below) (*Lockyer et al., 2003*; *Baguñà and Riutort, 2004*; *Laumer and Giribet, 2014*). Our concatenated analyses including all taxa (*Figure 1*) recover a novel topology, with Rhabdocoela as the earliest-diverging branch of Euneoophora, followed by another basal split between Proseriata and a clade composed of the remaining euneoophoran taxa. This topology is recovered by both ML under an unpartitioned LG4M+F model and BI under the highly site-heterogeneous CAT+GTR+Γ4 model (*Figure 1*). However, support for this topology differs across analytical modes, with the basal split between Rhabdocoela and the remaining Euneoophora showing maximum resampling support (pp) under BI (*Figure 1—figure supplement 2*), while support under ML (in both BMGE-trimmed and untrimmed matrices; *Figure 1*, *Figure 1—figure supplement 1* ) remains more marginal (bootstrap proportion 0.61 and 0.69, respectively). This is the only interordinal split among the free-living flatworms to be recovered with less than full resampling support in any of our concatenated analyses, perhaps a reflection of the short branch length estimated for this internode. It is also noteworthy that a second pair of Bayesian MCMCs, albeit with a slightly higher mean negative log likelihood, converged on a topology (not displayed) in which the proseriates, and not rhabdocoels, formed the earliest-diverging euneoophoran clade, again with full support. Support for the relative branching order of Proseriata and Rhabdocoela is also poor (0.54) across bootstrap replicates summarized by the ASTRAL species tree algorithm (*Figure 2*). Finally, experiments in which our representatives of either Neodermata or Bothrioplanida were deleted prior to tree inference (see below) also appear to influence relationships in this region of the phylogeny, with the former increasing support for the more derived position of Proseriata to 0.89 (*Figure 4*), and the latter causing the two orders to switch positions wholesale, with surprisingly high (0.96) bootstrap support for a later-branching position of Rhabdocoela (*Figure 5*). Altogether, therefore, while our results do agree in positioning both orders as early-branching members of Euneoophora, the relatively low support for and poor stability of these interrelationships casts doubt on the precise branching order of these taxa (*Figure 6*). Nonetheless, our analyses bring these orders closer together than they have generally been previously placed; hence, traits common to the two taxa (e.g., the synchronous mode of intracellular stylet formation [*Brüggemann, 1986*]) may under our topology be interpreted as plesiomorphies of Euneoophora. Further morphological comparisons between Rhabdocoela and Proseriata would, we note, be best conducted with reference to the poorly known order Gnosonesimida, which in our analyses represents the most proximate outgroup to Euneoophora.

The relative phylogenetic proximity of Proseriata and Rhabdocoela also casts the enigmatic genus *Ciliopharyniella* (*Ax, 1952*; *Sopott-Ehlers, 2001*), unfortunately not sampled here, in a particularly intriguing light. Currently classified as a (basal? [*Ehlers, 1972*]) rhabdocoel, but presenting characters of both Rhabdocoela (e.g., a rosulate pharynx) and Proseriata (elongate habitus with lateral, follicular female gonads in serial arrangement), as well as many apparent autapomorphies (*Sopott-Ehlers, 2001*), the original representative of this taxon, *Ciliopharyngiella intermedia Ax, 1952*, was introduced as demonstrating an 'intermediate' condition between

Rhabdocoela and Proseriata. Given the topological proximity of these taxa in our tree and the short branch separating them in our concatenated analyses (*Figure 1*), priority should be given to representing *Ciliopharyngiella* in future genome-scale phylogenies of Platyhelminthes, both to bring greater resolution to the question of the relative placements of Rhabdocoela and Proseriata, and to determine the status of *Ciliopharyngiella* as a relative of either lineage, or perhaps, as a distinct lineage in its own right.

## Adiaphanida is a strongly supported clade with no known morphological synapomorphies

Among the more surprising results of the era of rRNA-based platyhelminth phylogenetics was the complete dearth of molecular evidence for the higher taxon Seriata (*Baguñà et al., 2001*; *Joffe and Kornakova, 2001*; *Lockyer et al., 2003*; *Baguñà and Riutort, 2004*; *Laumer and Giribet, 2014*), encompassing the orders Tricladida, Proseriata, and Bothrioplanida (*Sopott-Ehlers, 1985*). This taxon was erected on the basis of the gross anatomical correspondence between these orders, which share a tricladoid gut (whether reticulating close behind the pharynx as in Proseriata and Bothrioplanida or not), a backwards-oriented, medially positioned plicate pharynx, and a follicular, repeated arrangement of vitellaria, frequently nested between gut diverticulae (also a trait of the aforementioned *Ciliopharyngiella*). Molecular phylogenetics, however, has split this taxon apart, mostly due to the ascent of the alternative Adiaphanida hypothesis—a clade uniting the orders Prolecithophora, Fecampiida, and Tricladida (*Norén and Jondelius, 2002*). Although entirely lacking in known morphological synapomorphies (aside, perhaps, from the eponymous opaque appearance of members of this clade), the rRNA support for this taxon has been almost unequivocal, and in light of our analyses (*Figures 1–5*), it would appear that this is also true for protein-coding genes. However, the internal relationships of Adiaphanida observed in our analyses differ from previous molecular phylogenies, showing a clade of Prolecithophora and Fecampiida (*Figure 1*). To date, the internal phylogeny of Adiaphanida has positioned Tricladida as the sister group of Prolecithophora in studies based on 18S rRNA (*Norén and Jondelius, 1999*; *Baguñà et al., 2001*; *Littlewood and Olson, 2001*), or as sister group to Fecampiida in studies based on both 18S and 28S rRNAs (*Lockyer et al., 2003*; *Laumer and Giribet, 2014*), although support for both topologies was generally poor. In contrast, our prolecithophoran-fecampiid clade is fully supported and in some respects would seem quite reasonable: indeed, several fecampiid genera (*Urastoma*, *Genostoma*) have been previously classified within Prolecithophora, due to gross anatomical features such as a common oral-genital pore reminiscent of some prolecithophoran taxa. However, we note that Prolecithophora and Fecampiida are also both among the most minimally sampled higher 'turbellarian' taxa in this analysis, and that our sampled fecampiid has the longest inferred branch length of any non-neodermatan flatworm; this result should therefore be regarded as provisional (*Figure 6*), pending further taxon sampling of both clades. If correct, however, this implies that the sister group of Tricladida, which contains important models in flatworm biology such as the planarian *Schmidtea mediterranea*, is equally closely related to two orders, rendering the taxon's origin marginally more ancient and somewhat complicating comparisons of, for example, regenerative mechanisms among Platyhelminthes.

## The interrelationships of free-living and parasitic platyhelminths

### The monospecific order Bothrioplanida is the closest living relative of the vertebrate-parasitic Neodermata

Among the strongest signals in this data set is the sister-group relationship of Neodermata with Bothrioplanida. Our sampled representative of this taxon, *Bothrioplana semperi*, is a freshwater 'microturbellarian', and is currently considered the only valid species of Bothrioplanida (*Sluys and Ball, 1985*). A scavenger and occasional predator of other small aquatic invertebrates, *B. semperi* is common in isolated, temporary water bodies such as vernal ponds, although it has also been collected in permanent lentic and lotic systems, wet terrestrial habitats (e.g., moss), and in phreatic and hyporheic biotopes (*Reisinger, 1925*; *Kahm, 1951*). Its distribution is cosmopolitan (*Tyler et al., 2012*), implying frequent long-distance dispersal, a process perhaps mediated by the production of durable resting eggs, the ability of adults to form resistant cysts in response to adverse environmental conditions, and its asexual propagation: the only known reproductive mode, reportedly, is a unique form of parthenogenesis known as dioogany (*Reisinger et al., 1974*), although a fraction (<10%, according to Reisinger; [*Reisinger, 1940*]) of mature individuals contain a rudimentary male system, reportedly incapable of

producing normal sperm. It possess a diploid (*Reisinger, 1940*; *Benazzi and Benazzi-Lentati, 1976*), albeit apparently very large (4.7 pg ≈ 4.6 Gb) genome (*Gregory et al., 2000*). Historically, *B. semperi* has been grouped with Tricladida and Proseriata in the taxon Seriata, due to the common presence in these taxa of a tricladoid gut, a backwards-oriented, medially positioned plicate pharynx, and a follicular, repeated arrangement of vitellaria nested between gut diverticulae. Despite this gross anatomical correspondence, however, Seriata enjoys essentially no molecular support, either in rRNA-based phylogenies (*Baguñà and Riutort, 2004*) or in the present work (see above). However, while rRNA phylogenies have been largely successful in providing alternative positions for Tricladida and Proseriata within Euneoophora, the phylogenetic position of *B. semperi* has remained elusive in such analyses to date (*Norén and Jondelius, 2002*; *Baguñà and Riutort, 2004*). Perhaps due to its relative obscurity and the widespread early acceptance of the Seriata concept, *B. semperi* was left unsampled in a number of prominent attempts to resolve the deep phylogeny of the phylum, including several specifically designed to identify the sister group of Neodermata (*Littlewood et al., 1999*; *Lockyer et al., 2003*; *Baguñà and Riutort, 2004*; *Littlewood, 2006*). Nonetheless, there may in fact exist phylogenetic signal for the common ancestry of Bothrioplanida and Neodermata in rRNA data as well: a recent mixture-model analysis (*Laumer and Giribet, 2014*) of a large rRNA data set recovered *B. semperi* as sister group to Neodermata (albeit with only modest support), echoing an earlier and little-recognized 18S rRNA-only result (*Baguñà et al., 2001*), and implying that this relationship, though unexpected, is not entirely unprecedented. In the present work, all concatenated analyses we performed recovered this clade with complete nodal support (*Figures 1, 4, 5*). This clade was also present in our ASTRAL species tree (*Figure 2*) with high bootstrap resampling support (88%), equal in magnitude to support for the monophyly of Neodermata. This relationship also appears to stand without substantial gene-tree conflict, at least gauging from the visual summary provided by our quartet supernetwork summary (*Figure 3*). Nevertheless: could this clade derive from a systematic error in phylogenetic inference, whereby unequivocal support for an incorrect topology is obtained by analyzing a large-scale data set such as the one presented here under a poorly specified evolutionary model (*Philippe et al., 2011*)?

Perhaps the most often considered source of phylogenetic error is a phenomenon commonly known as long-branch attraction, in which independent substitutions in unrelated fast-evolving lineages are erroneously construed as evidence of common ancestry by the chosen phylogeny reconstruction algorithm (*Parks and Goldman, 2014*). However, we find it difficult to explain Neodermata+Bothrioplanida as a long branch attraction artifact: for instance, if this topology were caused by attraction of Bothrioplanida to the long-branched Neodermata, one would expect to observe a long terminal branch in Bothrioplanida as well, when in fact this taxon shows among the shortest estimated root-to-tip branch lengths of any platyhelminth in our analysis (*Figure 1*). Furthermore, if Neodermata+Bothrioplanida were the result of an attraction artifact, one would expect to recover a different topology for Bothrioplanida in trees inferred in the absence of any representatives of Neodermata. However, when we perform this simple Neodermata-deletion experiment (*Figure 4*), we recover a relationship of *Bothrioplana* with Adiaphanida, which is the sister group of Bothrioplanida+Neodermata in our full-taxon analysis, falsifying this hypothesis of a long-branch attraction effect.

Heterotachy, another type of branch-length heterogeneity in which branch lengths vary across different sites (or genes) in an alignment, is also known to mislead phylogenetic analysis (*Philippe et al., 2005*; *Pagel and Meade, 2008*). This phenomenon is of especial concern in such large-scale analyses as presented here, as the practice of concatenation itself may introduce a degree of heterotachy into supermatrices. It may, for instance, be the case that there is one set of sites/genes in which Bothrioplanida is long-branched, and another set in which it is short-branched, effectively generating a 'long-branch' attraction despite a relatively slow estimated mean substitution rate. We can, however, find little evidence for this hypothesis. Analysis of both our unmodified and BMGE-trimmed matrices under phyML's 'integrated length' mode (see 'Materials and methods' for details), which permits each edge in the tree its own distribution of rates, effectively providing a simple model of heterotachy (*Guindon, 2013*), also recovers full support for a Neodermata+Bothrioplanida clade (*Figure 1*, *Figure 1—figure supplement 1*). Moreover, we note that our supernetwork and species-tree summaries of our individual gene tree analyses may account at least for that component of heterotachy introduced into the supermatrix by concatenation, in that branch lengths are independently fit for each gene. The final cause of systematic error we have investigated is compositional heterogeneity, whereby

the assumption of a single stationary amino-acid frequency vector is violated (*Foster, 2004*). Although the GC content of our transcriptomes varies substantially (*Supplementary file 1*), and such GC content variation is known to correlate strongly with amino acid frequency (*Moura et al., 2013*), strong support for Neodermata+Bothrioplanida is also recovered in matrices in which such amino-acid level compositional heterogeneity has been mitigated by trimming our alignment of sites that fail a test of non-stationarity (*Criscuolo and Gribaldo, 2010*). In sum, despite multiple tests designed to check for possible phylogeny reconstruction attraction artifacts, we cannot at present attribute the Neodermata+Bothrioplanida clade to any known cause of systematic error.

## Cestodes may be closely related to ectoparasites with a simple life cycle (Monogenea)

Understanding the evolutionary events that took place in the ancestors of Neodermata during their transition from free-living to parasitic habits also requires, beyond knowledge of their placement within the diversification of free-living Platyhelminthes, means to distinguish those characteristics of the diverse extant neodermatans which represent primitive traits from those which represent novelties acquired subsequent to the origin of the group (*Littlewood, 2006*). Was the neodermatan ancestor ecto- or endoparasitic? What taxon provided the original host species—or did the early neodermatans utilize several hosts in a complex life cycle, and if so, which taxa (invertebrate and vertebrate) formed the substrate for this cycle, and in which sequence? What was the environmental setting of these early associations with vertebrates, and in which geological era did crown-group neodermatans originally emerge and diversify? What morphological, developmental, and genomic adaptations may be considered common—but unique—to all Neodermata? While some of these questions may ultimately prove unanswerable, or may require information beyond that which may be attained through comparison of extant taxa (e.g., through paleontology; [*Upeniece, 2001*; *Dentzien-Dias et al., 2013*]), important constraints may be derived from a well-resolved internal phylogeny of Neodermata. Fundamental to this endeavor is establishing the monophyly of and interrelationships between the three major lineages (formerly classes) of Neodermata: Trematoda, Cestoda, and Monogenea.

Analyses employing morphological evidence appeared, at least initially, to provide sufficient evidence on many of these questions (*Llewellyn, 1965*; *Ehlers, 1985*; *Kearn, 1997*). The most widely held classical scenario relating these three taxa—the Cercomeromorpha (*Bychowsky, 1937*) hypothesis, critically reviewed by *Lockyer et al. (2003)*—posited a sister-group relationship between Monogenea and Cestoda. The single apomorphy uniting these taxa was considered to be the 'cercomer'—referring, at least in this specific (though not its original; [*Lockyer et al., 2003*]) context, to a hook-bearing posterior adhesive organ termed the opisthaptor in Monogenea, which corresponds remarkably in the number and morphology of its sclerotic hooks to posterior hook-bearing organs found in larval and some adult cestodes. (It is also noteworthy, however, that monogeneans and the early-branching cestode clade Gyrocotylidea [*Xylander, 2001*] are the only neodermatans to possess anterior nephridiopores.) Although this homology scheme has its critics even among morphologists (*Gulyaev, 1996*), the cercomer theory remains compelling in that it provides a rare, idiosyncratic link between two lineages otherwise so remarkably distinct in body plan and autecology (*Llewellyn, 1965*).

The era of molecular phylogenetics has upset this picture. In most analyses, the monophyly of Monogenea has been rejected (*Justine, 1998*), with many large rRNA-based analyses in favor of monogenean paraphyly, placing Polyopisthocotylea as the more basally branching lineage (*Littlewood et al., 1999*; *Littlewood and Olson, 2001*; *Laumer and Giribet, 2014*). However, an analysis of data from both ribosomal subunits sampled from all major lineages of Neodermata recovered support for a monophyletic Monogenea (*Lockyer et al., 2003*). This same study recovered Monogenea as the most basally branching clade of Neodermata, sister to a strongly supported clade of Cestoda and Trematoda. Nonetheless, a recent re-analysis of these same data recovered signal for Cercomeromorpha (*Laumer and Giribet, 2014*) with Cestoda nested within a paraphyletic Monogenea, reminiscent of earlier taxon-rich 18S rRNA analyses (*Littlewood et al., 1999*; *Littlewood and Olson, 2001*). This disagreement among rRNA-based analyses implies that signal for deep neodermatan interrelationships in these markers is sensitive to the mode of analysis and particularly alignment—perhaps an unsurprising observation, given the large insertions (*Giribet and Wheeler, 2001*) and rapid substitution rates characteristic of some neodermatan rRNAs. Molecular data from several independent sources have therefore been sought in pursuit of resolving these deep splits. Studies employing complete or nearly complete mitochondrial genomes have also found

strong support for a clade of Cestoda and Trematoda (*Park et al., 2007*), with further signal for monogenean paraphyly, though here manifested as paraphyly at the base of Neodermata and with Monopisthocotylea as the earliest-branching lineage (*Perkins et al., 2010*). However, no matter how strong the support values in such data sets, given the probable timescale of these deep neodermatan divergences (see above), mitochondrial genomes may provide less than ideal evidence towards these particular splits, given their widely noted problems such as the non-stationarity of nucleotide frequencies, their status as a single linkage group, and most remarkably, the more than fourfold higher substitution rate of platyhelminth mitochondrial genomes as compared to other Bilateria (*Bernt et al., 2013*), no doubt compounded by persistently poor sampling of data from free-living ougroups. It is in this context remarkable that the aforementioned mitogenomic analyses also yielded support for several results most would view with suspicion, such as, in one case, the paraphyly of Digenea (*Park et al., 2007*), or in another, paraphyly of not only Monogenea, but also Monopisthocotylea (*Perkins et al., 2010*). In light of these difficulties, the recent advent of a draft nuclear genome sequence from a monogenean, *Gyrodactylus salaris* (Gyrodactylidae: Monopisthocotylea), has been an important advance in bringing clarity to the basal splits in Neodermata (*Fromm et al., 2013*; *Hahn et al., 2014*). An analysis of the miRNA complement of *G. salaris*, compared with single exemplar species from Cestoda, Trematoda, and the free-living flatworms, was interpreted to support a clade of Cestoda and Trematoda (*Fromm et al., 2013*). However, although this study identified several novel taxon-specific miRNAs in each exemplar species, it failed to identify any novel miRNAs shared across two or more species: the synapomorphies proposed to link Cestoda and Trematoda were therefore taken to be the apparent absences (interpreted as losses) of four more broadly conserved miRNAs in the draft genomes of *Echinococcus granulosus* and *Schistosoma japonicum* (*Fromm et al., 2013*). Phylogenetic analysis of the gene models predicted from *G. salaris* including a sample of cestodes and trematodes nonetheless also recovered, with maximal nodal support, the early-branching position of *G. salaris*, although it is also noteworthy that the position of this basal split was observed to be in substantial gene-tree conflict, with the aforementioned Cestoda-Trematoda clade bearing an internode certainty (*Salichos et al., 2014*) of only 0.13 (*Hahn et al., 2014*). Altogether, however, it would seem that most published molecular data sets—based on rRNA, miRNAs, mitogenomics, and full-genome sequences—currently favor the sister-group relationship of Cestoda and Trematoda, despite the absence of any known morphological apomorphies of such a clade. Biologically, the simplest explanation of such a topology (particularly in the case of monogenean paraphyly as seen by *Perkins et al., 2010*) is that many monogenean traits, such as their ectoparasitic habits and their comparatively simple life cycles involving a single vertebrate host, are plesiomorphic to Neodermata; a corollary of this hypothesis posits a 'common origin of complex life cycles' (*Park et al., 2007*), that is, that the endoparasitic habits and utilization of invertebrate intermediate hosts in trematodes and cestodes (which use, however, different phyla) represent modifications of a life cycle inherited from their immediate common ancestor. Clearly, such a scenario would provide far-reaching constraints on the precise route by which neodermatans developed their parasitic habits.

In the present study, ML analysis of our unmodified supermatrix under the LG4M+F model also recovered a clade of Cestoda and Trematoda (*Figure 1—figure supplement 1*). However, nodal support for this clade was mediocre (0.74), in contrast to the full support recovered by *Hahn et al. (2014)*. This clade was also recovered with strong (0.97) bootstrap support in our ASTRAL species tree analysis (*Figure 2*). Remarkably, however, in our analyses of the untrimmed matrix employing BI under the site-heterogeneous CAT+GTR+Γ4 model, we observe a clade of Monogenea and Cestoda (Cercomeromorpha), inferred with maximal posterior probability (*Figure 1—figure supplement 2*). Cercomeromorpha was also recovered under ML analysis of our BMGE-trimmed matrix, with reasonably strong support (*Figure 1*). One may thus reasonably argue that Cercomeromorpha should be considered the better-supported hypothesis in our analyses (*Figure 1*), since it is preferred under the more site-heterogeneous model, as well as by analysis of a matrix constructed to remove sites that have the potential to mislead standard phylogenetic algorithms. This is, in any case, the first analysis of data from protein-coding genes to show support for the classical Cercomeromorpha hypothesis. New data must be collected from representatives of Polyopisthocotylea in order to give comment on the issue of the monophyly of Monogenea.

Fundamentally, resolving the branching order (and monophyly) of Monogenea, Trematoda, and Cestoda is a matter of discerning the position of the root of Neodermata, an issue familiar from

other 'hard' phylogenetic problems (*Giribet and Edgecombe, 2012*; *Zapata et al., 2014*). Accurate polarization of characters along this branch is dependent on appropriate outgroup comparison; a too-distant outgroup might in theory attract the long-branched *Gyrodactylus* to the base of Neodermata. In the analysis of *Hahn et al. (2014)*, the only available 'turbellarian' outgroup was the planarian *S. mediterranea*, a relatively derived triclad representing the more distant clade Adiaphanida. We therefore hypothesized that our recovery of Cercomeromorpha could have resulted from our having sampled a putatively more closely related outgroup to Neodermata (Bothrioplanida). However, reanalysis of a BMGE-trimmed matrix from which *B. semperi* was removed belies this notion: the best-sampled ML tree (in LG4M+F) fit to this matrix also recovers Cercomeromorpha, with a nodal support (0.72) comparable to the full-taxon analysis (*Figure 5*). We therefore conclude that the signal for Cercomeromorpha in the *G. salaris* genome recovered by our analyses rests on other aspects of data curation which differed between the present study and that of *Hahn et al. (2014)*, such as orthogroup selection or alignment and sequence masking. It is, finally, interesting to note that this *Bothrioplana* deletion experiment does influence other, more distant platyhelminth relationships: in this tree, Proseriata becomes the earliest-divergent taxon of Euneoophora, with Rhabdocoela as the sister to the remaining taxa, with strong (0.96) bootstrap support. The significance of this effect remains, at present, unclear.

## Implications for the origin of platyhelminth parasitism

Pre-cladistic classifications emphasized the separation of the parasitic flatworms from their free-living ancestors (the paraphyletic [*Ehlers, 1985*] 'Class Turbellaria'), in recognition of the vast phenetic differences between these lineages. Our identification of *B. semperi* as the closest free-living relative of Neodermata, and the nuclear genomic evidence we present for Cercomeromorpha, will help to narrow this artificial gap, by clarifying the relevant comparisons that should be made, and by setting taxonomic priorities for future research. Bothrioplanida and Neodermata may, for instance, bear evidence of common ancestry in aspects of their morphology: at the ultrastructural level, *B. semperi* resembles Neodermata both in the structure of its excretory system (namely, its protonephridial flame bulbs, which are composed of two cells with extensions that interdigitate to form an ultrafiltration weir in much the same way in both taxa [*Kornakova, 2010*]), as well as in the structure of its monociliary epidermal sensory receptors (which bear an electron-dense collar in both taxa [*Kornakova and Joffe, 1996*]). Further morphological investigation of the relatively obscure *B. semperi* may reveal other shared derived characters of these taxa, although certain character systems may prove elusive (e.g., spermatogenesis in a purportedly parthenogenetic species). Fortunately, knowledge of Bothrioplanida need not necessarily be restricted to *B. semperi*, given the evidence for at least one undescribed putative *Bothrioplana* species (*Kawakatsu and Mack-Firă, 1975*); further representatives may also be recovered by studies of cryptic molecular diversity and continued exploratory taxonomic surveys of freshwater microturbellarians, which remain poorly known outside of the Palearctic (*Artois et al., 2011*). Indeed, a new species of Bothrioplanida apparently capable of normal spermatogenesis has been recently reported from mainland China (*Ning et al., In press*).

Comparison between Bothrioplanida and the extant Neodermata can also extend beyond the search for synapomorphies: they may inform hypotheses on the route by which the earliest vertebrate-parasitic associations of Platyhelminthes arose. As a cosmopolitan species able to colonize temporary, chemically diverse, and spatially isolated freshwaters, *B. semperi* appears to be remarkably well adapted to frequent long-distance passive dispersal (perhaps via vertebrate, especially waterfowl, vectors [*Sluys and Ball, 1985*; *Artois et al., 2011*]), an ecological challenge at least analogous to the sweepstakes game that each succeeding generation of parasite plays during the colonization of a new host. It is therefore tempting to speculate that at least some adaptations to these similar ecological challenges could have been present in the most recent common ancestor of Bothrioplanida and Neodermata, and may have 'pre-adapted' early neodermatans to a parasitic lifestyle. For example, if stem Neodermata possessed a resistant, presumably quinone-tanned egg capsule similar to that used by *B. semperi* in passive dispersal, this could have facilitated enteric infection early in the history of this lineage (*Llewellyn, 1965*); indeed, extant freshwater microturbellarians produce resting eggs that are known to retain high viability subsequent to passage through vertebrate digestive tracts (*Artois et al., 2011*). Llewelyn's original formulation of this hypothesis posited that precocious emergence of the larvae of such swallowed eggs would provide a simple route to endoparasitism. Indeed, compared to their marine relatives, freshwater

flatworms are regularly exposed to a much wider range of variation in temperature, salinity, pH, and dissolved gas content (*Hutchinson, 1957*), facts which would seem to facilitate the colonization of an internal environment. This speculation hence implies that endoparasitism would be plesiomorphic to Neodermata, a possibility at least consistent with the evidence presented here for a relatively more derived position of Monogenea within Neodermata (Cercomeromorpha). One important counter-indication to this scenario, however, lies in the fact that most neodermatans, prior to the development of the neodermis, hatch as fully or partially ciliated larvae (miracidia or oncomiracida) which spend their earliest minutes-to-hours searching an aquatic medium for a new primary host in a functionally free-living dispersive mode. Definitive conclusions on the precise mode of parasitism employed by the common ancestor of extant Neodermata remain, in any case, premature, pending resolution of the mono- or non-monophyly of Monogenea.

Whatever the nature of the most recent common ancestor of Neodermata, it must be emphasized that the symbiosis presented by the neodermatan crown-group may be only dimly reflective of the form of symbiosis employed in its stem lineage. Traces of this earliest transition, may, moreover, be sparse, given the timescale of the divergence. While Bothrioplanida entirely lacks a fossil record, there are at least a few indications of the geological antiquity of crown Neodermata. The earliest direct fossil evidence of the clade is an assemblage of sclerotic hooks resembling those of Cercomeromorpha, recovered from lower Frasnian (~380 Mya) freshwater acanthodians and placoderms (*Upeniece, 2001*). However, if the suggestion of codivergence (notwithstanding subsequent host-switching events) with their gnathostome hosts in the deepest splits of several neodermatan clades (Cestoda [*Hoberg, 1999*] and both groups of Monogenea [*Jovelin and Justine, 2001*; *Bentz et al., 2003*]) were correct, then the diversification of Neodermata must precede that of crown-group gnathostomes in the midst of the Middle Cambrian (~525 Mya [*Blair and Hedges, 2005*]), implying that its common ancestor with Bothrioplanida has a still earlier origin. Thus, it must be noted that, despite the low amino acid substitution rate of Bothrioplanida, evolution and extinction have had considerable opportunity to erase the primitive organismic characteristics of this common ancestor in both descendant lineages, inherently constraining efforts to reconstruct the nature of this ancestor.

Nonetheless, it may be significant that the sister-taxon of Neodermata is a freshwater species. Although many aspects of the ecology of *B. semperi* have not been thoroughly studied, the species is not known to engage in any symbioses; its mode of dispersal is thought to be essentially passive. However, dispersal among disconnected habitats remains a fundamental challenge of all freshwater invertebrates, and one specific form of association, phoresis, has become a common adaptation to the necessity of dispersal in diverse groups (*Bilton et al., 2001*). Phoretic associations in freshwater invertebrates range from purely commensal to explicitly parasitic, with the life cycles of several higher taxa (Unionida, Hydrachnidia, Nematomorpha) including both a free-living adult phase and an ecto- or endoparasitic larval phase. We propose that a similar ecological mode may also have cha-racterized stem Neodermata prior to their transition to dedicated parasitism. This hypothesis presumes that Neodermata originally colonized freshwater or diadromous hosts. Given a well-sampled and well-resolved internal phylogeny of all Neodermata, and an explicit attempt at ancestral state reconstruction in host habitat, this suggestion could be straightforwardly tested. In this light, it is remarkable that many of the early-branching taxa within each major clade of Neodermata (e.g., lagotrematidae, Sundanonchidae, and Pseudomurraytrematidae in Monopisthocotylea [*Olson and Littlewood, 2002*; *Bentz et al., 2003*], Polystomatidae in Polyopisthocotylea [*Jovelin and Justine, 2001*], Amphilinidea, Caryophyllidea, and Diphyllobothriidea+Haplobothriidea in Cestoda [*Waeschenbach et al., 2012*], Aspidogastridae in Aspidogastrea [*Littlewood, 2006*], several higher taxa within the digenean clade Diplostomida [*Olson et al., 2003*]) are primarily or exclusively found in freshwater hosts (principally teleosts and amphibians).

To date, discussions of the emergence of platyhelminth parasitism have focused on organismic and morphological traits—in other words, those character systems for which data have been historically available. However, principally because of their importance as human pathogens, genomic data are now available from all major lineages of Neodermata, including well-curated assemblies, annotation efforts, and experimental protocols for species such as *Echinococcus multilocularis* (*Brehm, 2010*; *Olson et al., 2012*; *Tsai et al., 2013*) and *Schistosoma mansoni* (*Collins et al., 2013*; *Wang et al., 2013*). With such data available, there has been much discussion of the genome-level adaptations to parasitism, with suggestions of many apparent losses, including several homeobox

genes, *vasa*, *tudor*, and *piwi* orthologs, fatty and amino acid biosynthesis pathways, and peroxisome components; proposed gains include the evolution of a neodermatan-specific Argonaute subfamily and micro-exon gene organization (*Tsai et al., 2013*; *Hahn et al., 2014*). It is, however, essential to recognize that, in the absence of a well-founded platyhelminth phylogeny, associating any of these common genomic features to parasitism per se is not possible, as any of them may have a deeper, 'turbellarian' history. Discerning the molecular-level changes specifically associated with the origin of parasitism, therefore, requires comparison of neodermatan genome biology—initially, from the perspective of simple gene presence/absence, but eventually incorporating information on gene expression, function, regulation, and selection history—with the genome biology of their nearest free-living relative, as well as with more distant free-living flatworm orders. Fortunately, under the topology we have recovered, the distribution of model systems is nearly ideally suited to disentangle the molecular foundations of platyhelminth parasitism: only one taxon (indeed, a monospecific taxon) stands between Neodermata and Adiaphanida, within which triclads such as the powerful experimental system *S. mediterranea* (*Collins and Newmark, 2013*) are recovered as the earliest-branching lineage. Thus, if further evidence bears out the topology we have recovered here, we suggest that establishment of genomic resources and experimental protocols for *B. semperi* would thus provide the best available point of comparison to clearly understand the ecological, physiological, morphological, developmental, and genomic changes that took place during the single evolutionary transition that led to the most spectacular form of vertebrate parasitism in the modern world.

## Conclusions

Using a class of molecular sequence data fundamentally different in quality and much larger in quantity than the rRNA data sets that have been heretofore available, we have provided a modern hypothesis of 'deep' platyhelminth phylogeny (*Figure 6*), one which agrees in many respects with concepts from the eras of classical morphological and rRNA-based phylogenetics, but which also presents a number of unexpected relationships, not least of which is a novel scenario for origin of the vertebrate-parasitic Platyhelminthes. However, while several clades remain deserving of further investigation—in particular, (a) the interrelationships between Proseriata, Rhabdocoela, and the remaining Euneoophora, given the analytical instability surrounding these splits, and (b) the interrelationships within Adiaphanida, considering our sparse sampling of Fecampiida and Prolecithophora—most interrelationships in our phylogeny demonstrate remarkably high support and robustness across analytical modes. Thus, we would argue that, though this phylogeny certainly bears further testing using expanded taxon sampling, the most significant advancements in informing platyhelminth evolution henceforth will come not only from systematics, but particularly from those disciplines which take a known species tree as an input—for example, morphology, evolutionary developmental biology, and evolutionary genomics. It is our hope that such research will give due attention to the many poorly known lineages of 'microturbellaria' with which the most prominent members of this diverse phylum share their heritage.

## Materials and methods

### Specimen extraction, cDNA library construction, and sequencing

RNA was extracted from live single or pooled specimens, or specimens flash-frozen in TRIzol reagent or RNA*later* (Ambion, Inc., Carlsbad, CA). Polyadenylated mRNAs were either directly extracted using the Dynabeads DIRECT kit (using 'standard' or 'mini' scaling as recommended by the manufacturer, Life Technologies, Inc., Carlsbad, CA) or were purified from total RNA generated using a standard TRIzol extraction using the Dynabeads mRNA purification kit (Life Technologies, Inc). mRNA concentration was quantified and quality was assessed on an Agilent Bioanalyzer 2100 mRNA Pico kit; however, some successfully sequenced libraries prepared using the IntegenX mRNA kit were derived from mRNAs that were undetectably dilute in this protocol. cDNAs from the taxa sequenced on the MiSeq platform (see below) were produced by the phi29-mRNA amplification (PMA) method, amplifying autoligated cDNAs generated from poly-A selected mRNA as described by (*Pan et al., 2013*); Qiagen's Repli-G single cell kit was used to amplify these cDNAs. Nonstranded libraries were generated using the TruSeq RNA Sample Prep v2 kit (Illumina, Inc., San Diego, CA), using 5 µl mRNA per sample, and following the manufacturer's guidelines to fragment full-length cDNAs to a mean

insert size of 500 bp in a Covaris S220. PMA-amplified cDNAs were fragmented to a mean insert size of 800 bp and libraries were prepared following a modification of *Neiman et al. (2012)* in which dual indexing adapters (derived from the Illumina TruSeq DNA HT adapter sequences) were used in place of the original single-indexing adapters. Stranded libraries were generated using the IntegenX directional mRNA kit on the Apollo324 instrument, using up to 250 ng of mRNA per library (or the maximum value of 18 μl per library when mRNA quantity was undetectable). Details on specimen source and the library construction method used for each species are available in MCZBase (mczbase.mcz.harvard.edu) using the accession numbers provided in *Supplementary file 1*. Most cDNA libraries were completed with a single multiplexing index and sequenced with up to six libraries per lane; these libraries were read as normal 101 bp paired end runs on the Illumina HiSeq 2000 or as rapid runs on an Illumina HiSeq 2500 in the Harvard FAS Center for Systems Biology. cDNA libraries amplified using the PMA method (*Supplementary file 1*) were sequenced on an Illumina MiSeq as paired end 250 bp reads (v2 chemistry); read quality in these MiSeq runs was poor (only 48% reads with Phred ≥ 30), perhaps reflecting a combination of unusually large insert sizes and expired reagents.

## Sequence quality control, de novo transcriptome assembly, and ORF extraction

Raw reads were stringently demultiplexed by staff at the FAS Center for Systems Biology. Quality control was performed using the NGSQC Toolkit v2.3 (*Waeschenbach et al., 2012*) to maintain parity between reads during QC, retaining no ambiguous bases (AmbiguityFilter.pl -c set to 0), and trimming reads with a PHRED quality score threshold of 20 (TrimmingReads.pl -q set to 20), and using the IlluQC_PLL.pl script to remove primer/adapter sequences from a custom file constructed of all adapters/primers used in library construction. Libraries sequenced on the MiSeq platform were quality controlled using Trimmomatic v0.32 called with the following run parameters: ILLUMINACLIP:TruSeq3-PE.fa:2:30:10 LEADING:3 TRAILING:3 SLIDINGWINDOW:4:20 MINLEN:36. Following cleanup, paired reads were assembled in Trinity (*Grabherr et al., 2011*; *Haas et al., 2013*) (released 05 October 2012 or in the case of the PMA libraries, 14 April 2014), using the –RF flag when appropriate for stranded libraries (example command: Trinity.pl –seqType fq –left Mfus_R1_QC.fastq –right Mfus_R2_QC.fastq –SS_lib_type RF –CPU 6 –JM 150 G –bflyCPU 6 –output /n/Giribet_Lab/claumer/Mfus_trinity). The finished de novo assemblies were subjected to an initial round of redundancy reduction using CD-HIT-EST v3.1, clustering with a 5% global sequence similarity threshold, as a heuristic means of clustering transcripts expressed from the same putative locus but not clustered into the same subcomponent by Trinity. From this redundancy-reduced assembly, we extracted putative open reading frames (ORFs) using the transcripts_to_best_scoring_ORFs.pl script provided in the TransDecoder plugin to trinity. From the 'best_candidates.eclipsed_orfs_removed.pep' file, we used a custom Python script (choose_longest.py) to parse the fasta headers, and remove all putative ORFs but the longest per subcomponent. Statistics displayed in *Supplementary file 1* were computed using the TrinityStats.pl script distributed with Trinity v2014-04-13, and with the fastq-stats program from the ea-utils package v2014-04-22.

## Sanger EST data sets

For the species sampled using public Sanger EST data, we downloaded all available reads from the NCBI dbEST (52,772 for *Brachionus plicatilis*, 6729 for *Neobenedenia melleni*). Reads for each species were first sanitized using the SeqClean pipeline, removing all sequences with a hit against the NCBI UniVec database. These were then assembled using the TGI Clustering Tools (TGICL) package, with redundancy reduction and mitochondrial removal performed as described above. ORFs were predicted using the TransDecoder pipeline separately on 'contigs' and 'singlets' from the TGICL assemblies, and on the contigs, the longest ORF per cluster was retained using a modification of the aforementioned Python script. These ORFs from filtered contigs were then concatenated with singlet ORFs to generate a final peptide fasta file for use in OMA standalone.

## Public genome and transcriptome data

Predicted proteins from the *S. mansoni*, *E. multilocularis*, *Taenia solium*, and *Hymenolepis microstoma* genomes (*Berriman et al., 2009*; *Tsai et al., 2013*) were downloaded from GeneDB

(*Logan-Klumpler et al., 2011*). Predicted peptides from the draft *G. salaris* (v1.0) genome were downloaded from http://invitro.titan.uio.no/gyrodactylus/downloads.html. Protein predictions from the draft *Clonorchis sinensis* genome (*Wang et al., 2011*) (BioProject PRJDA72781), as well as from the genomes of *Lottia gigantea* (PRJNA175706), *Capitella teleta* (PRJNA175705), and *Helobdella robusta* (*Simakov et al., 2012*) (PRJNA175704) were downloaded from NCBI. RNA-Seq reads from *Fascioloides magna* (*Cantacessi et al., 2012*) were downloaded from SRA (accession SRX147910) and were subjected to the same pipeline as our newly generated RNA-Seq data sets. 454 cDNA reads from *Stylochoplana maculata* (*Struck et al., 2014*) were quality-trimmed using the fastx-toolkit (v0.0.13.2), trimming reads to a Phred score of 30 and discarding reads trimmed to shorter than 30 bp; these were assembled using Trinity (release 14 April 2014; see *Ren et al., 2012*). Prior to orthology prediction, all peptides containing nonsense characters (e.g., #) and intercalary stop codons were removed, and closely related peptides (e.g., isoforms) whose headers grouped them into a cluster were filtered using the retain-one-per-cluster Python script described above.

## Ortholog assignment and matrix construction

Predicted peptides derived from the workflows described above were assigned into orthologous groups using the graph-based OMA algorithm (*Roth et al., 2008*), implemented in OMA standalone v0.99x (http://omabrowser.org/standalone/), with the LengthTol parameter set to 0.61 and the MinSeqLen parameter set to 50. This yielded 86,808 OMA groups (putative orthologous groups by definition containing one sequence per species), of which 11,960 contained more than four species each (our criterion for potential inclusion in phylogenetic analysis). These were aligned using the L-INS-i algorithm implemented in MAFFT (*Katoh and Toh, 2008*) v6.853. Alignment certainty was quantified using the pair Hidden Markov Model framework implemented in ZORRO (*Wu et al., 2012*), using FastTree to derive guide trees for each alignment. A Python script was used to trim residues with an alignment uncertainty score below 0.5, and all alignments with a sequence-masked length greater than 50 AA (11,913) were considered for further analysis. We concatenated this set of genes using Phyutility v2.2.6 (*Smith and Dunn, 2008*) and utilized MARE v0.1.2-a (*Misof et al., 2013*) to select a subset of 'tree-like' genes for concatenation, using default settings, except for a taxon weighting ('-t') of 100, which we selected to ensure even representation among taxa. This procedure selected a subset of 516 orthogroups, which when concatenated yielded a supermatrix of 132,299 amino acid residues, of which 48.58% were occupied. We also constructed a trimmed version of this matrix using Block Mapping and Gathering with Entropy (BMGE) (*Criscuolo and Gribaldo 2010*) v1.1.1, as a means of removing aligned sites showing evidence of substitutional saturation and compositional heterogeneity. BMGE was called with the 'fast' test of compositional heterogeneity (-s FAST), calculating entropy scores against the BLOSUM30 matrix, and retaining all gaps (-g 1). This trimming yielded a matrix with 120,527 residues, of which 49.24% were occupied.

## Phylogenetic inference

Maximum likelihood inference for both the trimmed and untrimmed 516-ortholog supermatrix was conducted as a unpartitioned analysis under the best-fitting LG4M+F substitution model implemented in ExaML v1.0.0, with 100 bootstrap replicates, with the tree search set to begin from an initial tree ('-t') constructed using FastTree v2.1.7 (with the '–wag' and '–gamma' options selected). Bootstrap support for each clade in the ML tree was summarized using the sumtrees.py script in Dendropy v3.12 (*Sukumaran and Holder, 2010*).

To account for the effects of heterotachous substitution on phylogenetic inference, we also undertook maximum likelihood analyses using the 'integrated length' (–il) option implemented in phyML v20130927 to estimate branch lengths by integrating over geometric Brownian trajectories as described by *Guindon (2013)*. We performed this inference on both untrimmed and BMGE-trimmed versions of the concatenated matrix, using a combination of NNI and SPR heuristic searches, and inferring under the LG+F model with four 'free' (–free-rates) categories of rate variation. Branch support values were calculated as aBayes (Bayesian-like transformation of aLRT).

Bayesian phylogenetic analysis was carried out on the untrimmed 516-gene matrix in PhyloBayes-MPI v1.4e (*Lartillot et al., 2013*). We ran four independent chains under the

CAT+GTR+Γ4 model, removing constant sites using the -dc option, as a means of improving convergence as recommended in the PhyloBayes-MPI manual. Chains were run for 3554 to 4236 generations, until convergence on the posterior distribution between at least two chains appeared to have been achieved. We defined convergence as the point at which the maxdiff statistic from the bpcomp program (with a burnin of at least 3000 trees) fell below 0.1 for two or more chains. By this criterion, two pairs of the four chains appeared to mutually converge (with bpcomp showing a maxdiff of 0 with a burn-in of greater than 3300), but both chains in each pair remained divergent (maxdiff = 1) with the other two chains in the remaining pair. The negative log likelihood from the first pair of chains, however, showed a smaller average than that sampled by the second pair (1.878634E + 06 vs 1.880144E + 06).

Individual gene trees were inferred in RAxML 8.0.2, using the AUTO function to select the optimal substitution matrix, with empirical amino acid frequencies ('-m PROTGAMMAAUTOF'). For each of the 516 genes selected by MARE for supermatrix construction, trees were created for 100 rapid bootstrap replicates, with branch length optimization specified for these bootstraps ('-k'), and the final ML tree search was conducted using these bootstrap trees as starting trees ('-f a').

## Quartet supernetwork and ASTRAL species tree construction

We constructed quartet supernetworks to summarize phylogenetic conflicts within our set of 516 best-found ML trees visually using SuperQ (*Grunewald et al., 2013*) v1.1, which permits partial gene trees as input. This is noteworthy as many other approaches of visualizing or quantifying among gene tree conflict (including the available RAxML implementation of internode certainty [*Salichos et al., 2014*]) require input gene trees to be complete in all taxa. We constructed supernetworks using the balanced linear secondary objective, imposing no filter. We also constructed supernetworks using identical settings, taking as input bootstrap majority rule consensus trees constructed using the sumtrees.py program of DendroPy python library (*Sukumaran and Holder, 2010*), v3.12. This supernetwork appeared qualitatively very similar to the ML-tree quartet supernetwork and is hence not displayed. Supernetworks were visualized in SplitsTree v4 using default settings.

We used the ASTRAL species tree algorithm (*Mirarab et al., 2014*), which, like SuperQ, decomposes input gene trees into quartets, but which finds for these input quartets an optimal, fully bifurcating species tree. For this reason ASTRAL also operates efficiently on large sets of incomplete, unrooted gene trees, which remains a limitation of other currently available species tree methods. ASTRAL was run using default settings, given as input all 516 best-found ML trees; we also calculated bootstrap species trees using the bootstrapped gene trees, bootstrapping only at the site level (i.e., not at the site-and-gene level, [*Seo, 2008*]).

## Counts of decisive and congruent gene sets

To quantify the number of genes potentially decisive for a given split, and also the number within this set that actually support the split in question (*Figure 1*), we employed a custom Python script (parse_gene_trees.py), built using the ETE2 library (*Huerta-Cepas et al., 2010*), to parse the set of individual ML gene trees (see also *Fernández et al., 2014*). If a tree contained at least one species in both descendant clusters for a given clade, plus at least two distinct species basal to the node in question, it was considered potentially decisive (forming minimally a quartet; [*Dell'Ampio et al., 2013*]); if the descendants were monophyletic with respect to their relative outgroups, that gene tree was considered congruent with the node in question (although this count is agnostic to the topology within the node in question). We displayed these counts for all nodes reflecting interordinal relationships in the 516-gene, BMGE-trimmed supermatrix ML topology (*Figure 1*).

## Data deposition

Original sequence reads and have been deposited to SRA accessible at NCBI (http://www.ncbi.nlm.nih.gov/bioproject/) via the BioProject IDs provided in *Supplementary file 1*. Scripts used in this study are available at https://github.com/claumer. Concatenated and individual orthogroup amino acid alignments, all inferred trees, and original Trinity transcriptome assemblies used in this study are available at Data Dryad (doi:10.5061/dryad.622q4; *Laumer et al., 2015b*).

## Acknowledgements

We thank Alejandro Sanchéz Alvarado and Eric Ross for sharing unpublished RNA-Seq data from *Schmidtea mediterranea*, and Jochen Rink for sharing the de novo assembly of *Dendrocoelum lacteum*. Janine Caira consented to have her line drawings reproduced under a CC BY license for *Figure 6*, and is also responsible for inspiring the taxon-deletion experiments. Julian Smith III provided critical commentary on the morphological codings presented in *Figure 6*. Bruce Conn and Mansi Srivastava gave comments on the manuscript. Kate Rawlinson, Chris Lowe, Kevin Uhlinger, Sal Genovese, Ted Lyman, Paul Davison, Kevin Pang, Maja Adamska, and numerous others helped collect specimens. Alexis Criscuolo and Sarah Bastkowski provided minor modifications or advice on the usage of BMGE and SuperQ, respectively. Claire Reardon and Christian Daly at the Harvard FAS Center for Systems Biology provided essential library preparation and sequencing support, and provided two free MiSeq runs. Staff at the Harvard FAS Division of Science, Research Computing group are thanked for computational support on the Odyssey 2 research cluster. This research was supported by internal funds from the Museum of Comparative Zoology to GG, by an NSF DDIG (Award Number 1210328) to CL and GG, by Society for Integrative and Comparative Biology and Sigma Xi Grants-in-Aid to CL and from the core budget of the Sars Centre to AH.

## Additional information

### Funding

| Funder | Grant reference | Author |
| --- | --- | --- |
| National Science Foundation (NSF) | DDIG, 1210328 | Christopher E Laumer, Gonzalo Giribet |
| Cornell University | Sigma Xi Grant in Aid of Research | Christopher E Laumer |
| Society for Integrative and Comparative Biology | Grant in Aid of Research | Christopher E Laumer |
| Harvard University | Museum of Comparative Zoology | Gonzalo Giribet |
| Universitetet i Bergen | Sars Centre | Andreas Hejnol |

The funders had no role in study design, data collection and interpretation, or the decision to submit the work for publication.

### Author contributions
CEL, Conception and design, Acquisition of data, Analysis and interpretation of data, Drafting or revising the article; AH, Drafting or revising the article, Contributed unpublished essential data or reagents; GG, Conception and design, Analysis and interpretation of data, Drafting or revising the article

### Author ORCIDs
Christopher E Laumer, http://orcid.org/0000-0001-8097-8516
Andreas Hejnol, http://orcid.org/0000-0003-2196-8507

## Additional files

### Supplementary file
• Supplementary file 1. Summary statistics and accession numbers for each species used in this study. Summarizes quality-controlled RNA-Seq data, de novo transcriptome assemblies, peptide predictions (incl. those from published genome data), and gene occupancy in the 516-orthogroup matrix. Libraries marked with a single asterisk represent stranded libraries made using the IntegenX mRNA sample prep kit, whereas those marked with two asterices represent libraries prepared from amplified

cDNA made using the PMA method (*Pan et al., 2013*); all other libraries were produced using the Illumina TruSeq RNA Sample Prep Kit v2.

## Major datasets

The following datasets were generated:

| Author(s) | Year | Dataset title | Dataset ID and/or URL | Database, license, and accessibility information |
|---|---|---|---|---|
| Laumer CE, Hejnol A, and Giribet G | 2015 | Data from: Nuclear genomic signals of the 'microturbellarian' roots of platyhelminth evolutionary innovation | 10.5061/dryad.622q4 | Available at Dryad Digital Repository under a CC0 Public Domain Dedication. |
| Laumer CE, Hejnol A, Giribet G | 2015 | Stenostomum leucops | http://www.ncbi.nlm.nih.gov/bioproject/?term=PRJNA276469 | Publicly available at the NCBI BioProject database (PRJNA276469). |
| Laumer CE, Hejnol A, Giribet G | 2015 | Stylochus ellipticus | http://www.ncbi.nlm.nih.gov/bioproject/?term=PRJNA277586 | Publicly available at the NCBI BioProject database (PRJNA277586). |
| Laumer CE, Hejnol A, Giribet G | 2015 | Xenoprorhynchus sp. I CEL-2015 | http://www.ncbi.nlm.nih.gov/bioproject/?term=PRJNA277588 | Publicly available at the NCBI BioProject database (PRJNA277588). |
| Laumer CE, Hejnol A, Giribet G | 2015 | Solenopharyngidae sp. CEL-2015 | http://www.ncbi.nlm.nih.gov/bioproject/?term=PRJNA277592 | Publicly available at the NCBI BioProject database (PRJNA277592). |
| Laumer CE, Hejnol A, Giribet G | 2015 | Rhynchomesostoma rostratum | http://www.ncbi.nlm.nih.gov/bioproject/?term=PRJNA277594 | Publicly available at the NCBI BioProject database (PRJNA277594). |
| Laumer CE, Hejnol A, Giribet G | 2015 | Provortex cf. sphagnorum CEL-2015 | http://www.ncbi.nlm.nih.gov/bioproject/?term=PRJNA277595 | Publicly available at the NCBI BioProject database (PRJNA277595). |
| Laumer CE, Hejnol A, Giribet G | 2015 | Protomonotresidae sp. CEL-2015 | http://www.ncbi.nlm.nih.gov/bioproject/?term=PRJNA277596 | Publicly available at the NCBI BioProject database (PRJNA277596). |
| Laumer CE, Hejnol A, Giribet G | 2015 | Prosogynopora riseri | http://www.ncbi.nlm.nih.gov/bioproject/?term=PRJNA277597 | Publicly available at the NCBI BioProject database (PRJNA277597). |
| Laumer CE, Hejnol A, Giribet G | 2015 | Prorhynchus sp. I CEL-2015 | http://www.ncbi.nlm.nih.gov/bioproject/?term=PRJNA277598 | Publicly available at the NCBI BioProject database (PRJNA277598). |
| Laumer CE, Hejnol A, Giribet G | 2015 | Prorhynchus alpinus | http://www.ncbi.nlm.nih.gov/bioproject/?term=PRJNA277599 | Publicly available at the NCBI BioProject database (PRJNA277599). |
| Laumer CE, Hejnol A, Giribet G | 2015 | Monocelis fusca | http://www.ncbi.nlm.nih.gov/bioproject/?term=PRJNA277600 | Publicly available at the NCBI BioProject database (PRJNA277600). |
| Laumer CE, Hejnol A, Giribet G | 2015 | Microstomum lineare | http://www.ncbi.nlm.nih.gov/bioproject/?term=PRJNA277601 | Publicly available at the NCBI BioProject database (PRJNA277601). |

| Author(s) | Year | Dataset title | Dataset ID and/or URL | Database, license, and accessibility information |
|---|---|---|---|---|
| Laumer CE, Hejnol A, Giribet G | 2015 | Microdalyellia sp. CEL-2015 | http://www.ncbi.nlm.nih.gov/bioproject/?term=PRJNA277602 | Publicly available at the NCBI BioProject database (PRJNA277602). |
| Laumer CE, Hejnol A, Giribet G | 2015 | Lehardyia sp. CEL-2015 | http://www.ncbi.nlm.nih.gov/bioproject/?term=PRJNA277603 | Publicly available at the NCBI BioProject database (PRJNA277603). |
| Laumer CE, Hejnol A, Giribet G | 2015 | Macrostomum cf. ruebushi CEL-2015 | http://www.ncbi.nlm.nih.gov/bioproject/?term=PRJNA277604 | Publicly available at the NCBI BioProject database (PRJNA277604). |
| Laumer CE, Hejnol A, Giribet G | 2015 | Bothriomolus balticus | http://www.ncbi.nlm.nih.gov/bioproject/?term=PRJNA277605 | Publicly available at the NCBI BioProject database (PRJNA277605). |
| Laumer CE, Hejnol A, Giribet G | 2015 | Bdelloura candida | http://www.ncbi.nlm.nih.gov/bioproject/?term=PRJNA277606 | Publicly available at the NCBI BioProject database (PRJNA277606). |
| Laumer CE, Hejnol A, Giribet G | 2015 | Lepadella patella | http://www.ncbi.nlm.nih.gov/bioproject/?term=PRJNA277607 | Publicly available at the NCBI BioProject database (PRJNA277607). |
| Laumer CE, Hejnol A, Giribet G | 2015 | Kronborgia cf. amphipodicola CEL-2015 | http://www.ncbi.nlm.nih.gov/bioproject/?term=PRJNA277608 | Publicly available at the NCBI BioProject database (PRJNA277608). |
| Laumer CE, Hejnol A, Giribet G | 2015 | Austrognathia sp. CEL-2015 | http://www.ncbi.nlm.nih.gov/bioproject/?term=PRJNA277630 | Publicly available at the NCBI BioProject database (PRJNA277630). |
| Laumer CE, Hejnol A, Giribet G | 2015 | Bothrioplana semperi | http://www.ncbi.nlm.nih.gov/bioproject/?term=PRJNA277632 | Publicly available at the NCBI BioProject database (PRJNA277632). |
| Laumer CE, Hejnol A, Giribet G | 2015 | Geocentrophora applanata | http://www.ncbi.nlm.nih.gov/bioproject/?term=PRJNA277633 | Publicly available at the NCBI BioProject database (PRJNA277633). |
| Laumer CE, Hejnol A, Giribet G | 2015 | Gnosonesimida sp. IV CEL-2015 | http://www.ncbi.nlm.nih.gov/bioproject/?term=PRJNA277634 | Publicly available at the NCBI BioProject database (PRJNA277634). |
| Laumer CE, Hejnol A, Giribet G | 2015 | Prostheceraeus vittatus | http://www.ncbi.nlm.nih.gov/bioproject/?term=PRJNA277637 | Publicly available at the NCBI BioProject database (PRJNA277637). |

| Author(s) | Year | Dataset title | Dataset ID and/or URL | Database, license, and accessibility information |
|---|---|---|---|---|
| Laumer CE, Hejnol A, Giribet G | 2015 | Lepidodermella sp | http://www.ncbi.nlm.nih.gov/bioproject/?term=PRJNA277639 | Publicly available at the NCBI BioProject database (PRJNA277639). |

The following previously published datasets were used:

| Author(s) | Year | Dataset title | Dataset ID and/or URL | Database, license, and accessibility information |
|---|---|---|---|---|
| Liu S-Y, Selck C, Friedrich B, Lutz R, Vila-Farré M, Dahl A, Brandl H, Lakshmanaperumal N, Henry I, Rink JC | 2013 | Dendrocoelum lacteum | http://www.ncbi.nlm.nih.gov/bioproject/?term=PRJNA207765 | Publicly available at the NCBI BioProject database (PRJNA207765). |
| Cantacessi C, Mulvenna J, Young ND, Kasny M, Horak P, Aziz A, Hofmann A, Loukas A, Gasser RB | 2012 | Paired-end Illumina reads generated from F. magna cDNA | http://www.ncbi.nlm.nih.gov/sra/?term=SAMN00993415 | Publicly available at the NCBI Sequence Read Archive (SAMN00993415). |
| Struck TH, Wey-Fabrizius AR, Golombek A, Hering L, Weigert A, Bleidorn C, Klebow S, Iakovenko N, Hausdorf B, Petersen M, Kück P, Herlyn H, Hankeln T | 2014 | Stylochoplana maculata Adult transcriptome | http://www.ncbi.nlm.nih.gov/sra/?term=SAMN02739901 | Publicly available at the NCBI Sequence Read Archive (SAMN02739901). |

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
