## [Decision Letter]

Thank you for sending your work entitled “Nuclear genomic signals of the ‘microturbellarian’ roots of platyhelminth evolutionary innovation” for consideration at *eLife*. Your article has been favorably evaluated by Diethard Tautz (Senior editor) and 2 reviewers, one of whom, Alejandro Sánchez Alvarado, is a member of our Board of Reviewing Editors.

The Reviewing editor and the other reviewer discussed their comments before we reached this decision, and the Reviewing editor has assembled the following comments to help you prepare a revised submission.

The authors report on a comprehensive analysis of phyletic relationships among the true flatworms (Platyhelminthes) using a survey of genomes and transcriptomes representing all free-living flatworm orders. This work is the first of its type for the Platyhelminthes and ultimately provides a modern hypothesis of flatworm phylogeny that should help inform inter-relationships between these remarkably diverse group of animals. This is an important contribution to the field that will be of great interest to those studying the evolution of animal development and macroevolution in general, and the organisms and the diseases they cause in particular.

By comparing hundreds of nuclear protein coding genes, the authors were able to derive a phylogeny with at least two key novel and intriguing attributes: 1) a primacy of “microturbellarian” groups that illuminates key evolutionary transitions within the platyhelminthes; and 2) a novel scenario for the interrelationships between free-living and parasitic flatworms that provides unique opportunities to shed light on the origins and biological consequences of parasitism in these animals.

Perhaps the single most important contribution of the present body of work is to provide strong phylogenetic evidence for the importance of microturbellarians in the evolution of Platyhelminthes. These organisms have not captured the attention of researchers like the best-known branches of the clearly much larger and deeper phylogenetic diversity of flatworms (e.g., planarians, polyclads, and neodermatans). It is our suspicion that this paper will bring an end to their relative obscurity.

Another important point worth mentioning is the tree's internal relationships of Adiaphanida in which a clade of Prolecithophora and Fecampiida are shown and which differs from previous molecular phylogenies. Although this is an extremely undersampled group (and provided some of the longest branches in the present study), if true, it implies that the sister group of the Tricladida containing the important model system *Schmidtea mediterranea*, is evolutionarily equidistant to two orders, which will complicate comparisons of regenerative properties among these animals.

Finally, the phylogenetic analyses reported here, indicates that a monospecific taxon (*Bothrioplana semperi*) separates Neodermata and Adiaphanida, which contains the earliest branching lineage from this taxon (Tricladida), and may hold the secret to understanding the evolutionary origins of parasitism in this group of animals.

In sum, the work reported by Laumer et al. makes a very strong argument for the need to leave behind disciplines which only yield phylogenies as outputs (e.g., systematics), and embrace instead those which dissect established species trees via morphological, evolutionary developmental biology, and evolutionary genomics.

Overall, this manuscript deserves a broad readership. As such, we encourage the authors to address the main criticism that follows.

An essential revision we wish to see is that, as the number of traits discussed by the authors is quite large, a “model” figure that maps (at least some of) the traits on the phylogeny would help with the interpretation and understanding of the deep phylogeny presented. Such a figure would nicely summarize the authors' key findings, and we would anticipate its broad used in zoology and Eco/Evo/Devo classrooms.

---

## [Author Response]

*An essential revision we wish to see is that, as the number of traits discussed by the authors is quite large, a “model” figure that maps (at least some of) the traits on the phylogeny would help with the interpretation and understanding of the deep phylogeny presented. Such a figure would nicely summarize the authors' key findings, and we would anticipate its broad used in zoology and Eco/Evo/Devo classrooms*.

This is an excellent suggestion. We have newly added Figure 6 to the article, which we feel summarizes in a single visualization the strongest and weakest aspects of our analyses, and which maps along this phylogeny the traits discussed explicitly in the text, as well as several other basic biological (relative diversity, habitat) and phylogenetically important traits of the microturbellarian orders we have sampled. It is our hope this figure will serve as a useful introduction to the diversity which is the focus of this paper, both to expert readers from other fields, and to new students of zoology and Eco/Evo/Devo.